# A Stochastic Approach to the Subset Selection Problem via Mirror Descent

Dan Greenstein[1]   Elazar Gershuni[2]   Ilan Ben-Bassat[3]   Yaroslav Fyodorov[4]   Ran Moshe[5]
Fiana Raiber[1]   Alex Shtoff[4]   Oren Somekh[4]   Nadav Hallak[1]

[1] Technion Israel Institute of Technology, Faculty of Data and Decision Sciences
[2] Technion Israel Institute of Technology, Faculty of Computer Science
[3] Yahoo Research [4] Technology Innovation Institute [5] Private Consultant
sdngreen@campus.technion.ac.il, elazarg@gmail.com,
ilan.benbassat@yahooinc.com, yaroslav.fyodorov@tii.ae,
rani.moshe@gmail.com, fiana@technion.ac.il, alexander.shtoff@tii.ae,
oren.somekh@tii.ae, ndvhllk@technion.ac.il

## Abstract

The subset selection problem is fundamental in machine learning and other fields of computer science. We introduce a stochastic formulation for the minimum cost subset selection problem in a black box setting, in which only the subset metric value is available. Subsequently, we can handle two-stage schemes, with an outer subset-selection component and an inner subset cost evaluation component. We propose formulating the subset selection problem in a stochastic manner by choosing subsets at random from a distribution whose parameters are learned. Two stochastic formulations are proposed. The first explicitly restricts the subset's cardinality, and the second yields the desired cardinality in expectation. The distribution is parameterized by a decision variable, which we optimize using Stochastic Mirror Descent. Our choice of distributions yields constructive closed-form unbiased stochastic gradient formulas and convergence guarantees, including a rate with favorable dependency on the problem parameters. Empirical evaluation of selecting a subset of layers in transfer learning complements our theoretical findings and demonstrates the potential benefits of our approach.

## 1 Introduction

This paper proposes a stochastic optimization approach to the *Subset Selection Problem* in which the goal is to choose a subset of size $k$ that attains the minimal loss defined by

$$\min_{C \in \mathcal{C}^k} \ell(C), \tag{P}$$

where $\mathcal{C}^k := \{C \subseteq \{1, \ldots, n\}, |C| = k\}$, $\mathcal{C} = \bigcup_{k=1}^{n} \mathcal{C}^k$, and $\ell : \mathcal{C} \to \mathbb{R}$ is a set loss function. We make *no assumptions* regarding the loss function other than having known lower and upper bounds. In particular, we neither assume it is defined outside of $\mathcal{C}$ nor require it to be differentiable, unlike previous related research Ahmed et al. (2022); Pervez et al. (2022); Sander et al. (2023); Xie and Ermon (2019). As a result, our formulation can accommodate loss functions that others cannot, such as those arising in feature selection or transfer learning via fine-tuning a subset of neural layers. However, it is important to note that while our convergence results hold for arbitrary $\ell$, the properties of $\ell$ significantly influence both the convergence rate and the algorithm's practical performance.

Central in many fields of science, such as machine learning and theoretical computer science, the subset selection problem is NP-hard (see Bar-Yehuda and Even (1981)). In this paper, we propose

---

Research partly conducted during Dan and Elazar's internships at Yahoo Research.
Research conducted while Yaroslav, Ran, Fiana, Alex and Oren worked at Yahoo Research.

two distributional-based variants for (P) – both optimize distribution parameters to minimize the expected subset loss. The size of the subsets in the first variant is $k$ (surely), while in the second, it is $k$ in expectation. We prove that our algorithm converges to a stationary point for both formulations. To the best of our knowledge, this is the first proof that does not require the differentiability of the loss function $\ell$.

Both formulations are of the general form

$$\inf_{\boldsymbol{w}\in\mathcal{X}}\left\{f^D\left(\boldsymbol{w}\right):=\mathbb{E}_{C\sim D_{\boldsymbol{w}}}\left[\ell(C)\right]\right\}, \tag{1.1}$$

where $D_{\boldsymbol{w}}$ is a distribution on subsets parameterized by $\boldsymbol{w}$, $\ell$ is a loss function associated with the subsets, $f^D$ is the expected loss, and $\mathcal{X}\subseteq\mathbb{R}^n$.

The first formulation we consider limits the size of the subset to be exactly $k$ via the distribution $D_{\boldsymbol{w}}=\psi_{\boldsymbol{w}}$, and is intended for cases where $k=O(1)$. It is formulated by

$$\inf_{\boldsymbol{w}\in\Delta_n^0}\left\{f^\psi\left(\boldsymbol{w}\right):=\mathbb{E}_{C\sim\psi_{\boldsymbol{w}}}\left[\ell(C)\right]\right\}, \tag{$P_k$}$$

where $\Delta_n^c:=\left\{\boldsymbol{w}\in\Delta_n\mid\boldsymbol{w}_i\geq c,\ \ w_i>0\ \ \forall i\right\}$ for $0\leq c<1$, $\Delta_n$ is the $n$-dimensional unit simplex, and $\psi_{\boldsymbol{w}}$ is an unordered choice without replacement according to weights $\boldsymbol{w}$. Note that this is a distribution over subsets of size $k$.

**Remark 1.1** (on the closeness of the feasible set of ($P_k$))**.** The feasible set in ($P_k$) is not closed – this hindrance is addressed in the theoretical analysis of the formulation in Section 5.

The second formulation we consider allows different sizes of the selected subsets via a random Bernoulli-based mechanism where the expected subset size is $k$. It is defined by

$$\min_{\boldsymbol{w}\in\Delta_{n,k}^0}\left\{f^\phi\left(\boldsymbol{w}\right):=\mathbb{E}_{C\sim\phi_{\boldsymbol{w}}}\left[\ell(C)\right]\right\}, \tag{$P_B$}$$

where $\Delta_{n,k}^c:=\left\{\boldsymbol{w}\in\mathbb{R}^n\mid\boldsymbol{w}\in[c,1-c]^n,\sum_{j=1}^n\boldsymbol{w}_j=k\right\}$ for $0\leq c<(n-k)^{-1}$. In this formulation, $\phi_{\boldsymbol{w}}$ is the result of $n$ heterogeneous Bernoulli trials, where the $i^{th}$ element is included in the subset if the $i^{th}$ trial is successful, and the individual element inclusion probabilities sum up to $k$.

The choice between fixed ($P_k$) and expected ($P_B$) cardinality depends on whether cardinality is a strict requirement (e.g., Subset Sum) or a guideline (e.g., feature selection). While expected cardinality can approximate fixed cardinality by penalizing subsets of undesired sizes, this may hinder convergence due to frequent low-quality samples.

The detailed distributions are given in Section 4, and the purpose of the *approximation parameter* $c$ will become evident in Section 5. In both formulations, ($P_k$) and ($P_B$), selecting a subset boils down to determining a distribution parameterized by the decision variable $\boldsymbol{w}$. To determine $\boldsymbol{w}$, we utilize a nonconvex Stochastic Mirror-Descent (SMD) Zhang and He (2018) based method detailed in Algorithm 1. Accordingly, we measure the first-order optimality of a solution $\boldsymbol{w}$ using the standard Bregman stationarity measure Zhang and He (2018); we elaborate on such preliminaries in Section 2.

**Contribution.** We briefly summarize the main contributions of this work:

- We propose two continuous stochastic formulations for the Subset Selection Problem with possibly *discrete* loss, and prove that they can encode the optimal selection. Moreover, by choosing appropriate distributions for these formulations, we derive constructive gradient estimators, and the relatively weak convexity of the expected loss functions.
- We prove convergence rate via an SMD-based method, without any assumption on the underlying loss. We are not aware of such a proof even for differentiable loss functions.
- We derive a concentration bound for the loss of the subsets sampled using our method.
- We demonstrate the efficacy of the proposed algorithm in a deep transfer learning setting.

**Related Work.** Recent research on stochastic subset selection emphasizes its integration into computation graph (CG) frameworks, widely used in modern machine learning applications Sander et al. (2023); Xie and Ermon (2019); Jang et al. (2016); Maddison et al. (2016); Ahmed et al.

(2022); Pervez et al. (2022). These methods aim to compute gradient estimators for subset selection parameters by encoding subsets as $k$-hot vectors—binary vectors with $k$ non-zero entries representing the chosen subsets. While CG frameworks assume the loss is differentiable almost everywhere, we address discrete loss functions where this assumption fails, as with metrics like classification accuracy or Spearman correlation.

Sander et al. (2023) provide theoretical guarantees for their gradient estimator, introducing a generalized top-$k$ operator with a subgradient and a relaxed version with an exact gradient. Their approach, however, does not apply to non-differentiable losses.

The Gumbel-Softmax trick, a relaxation of top-$k$ sampling Xie and Ermon (2019), is suited for differentiable losses. In our setting, it has two limitations: (i) defining loss for continuous relaxations of $k$-hot vectors is challenging for tasks like feature selection, and (ii) the relaxation depends on a temperature parameter $t$. While the relaxed distribution converges to weighted size-$k$ subset sampling as $t \to 0$, the distributions differ for any fixed $t > 0$.

Other notable approaches for stochastic subset sampling in CG settings include those by Ahmed et al. (2022) and Pervez et al. (2022). Ahmed et al. (2022) investigate a Poisson-Binomial-like distribution conditioned on selecting exactly $k$ elements. They propose dynamic programming algorithms to compute marginal selection probabilities and enable sampling, along with a theoretically intuitive gradient estimator based on these probabilities, though it lacks rigorous proof.

Pervez et al. (2022) propose a sampling mechanism that reduces subset size variance in the Poisson-Binomial distribution. Using Bernoulli random variables, they generate subsets near size $k$ through sequential trials and apply the Straight-Through gradient estimator Bengio et al. (2013), though without guarantees on bias or variance. While their method is primarily relevant to our work from a practical perspective, combining it with ours could yield synergistic and innovative approaches, as discussed in Appendix F.

Outside the CG framework, the REINFORCE gradient estimator Williams (1992) does not require differentiable losses. While provably unbiased, it has high variance, and we are unaware of theoretical bounds on its variance or moments, which are crucial for first-order stochastic optimization convergence.

Our work intersects with stochastic smoothing techniques, smoothing for set-valued functions, and variational optimization. For foundational works, see Lovász (1983) for deterministic smoothing of set-valued functions, Nesterov (2005) for smoothing in nonsmooth continuous functions, Duchi et al. (2012) for stochastic smoothing, and Staines and Barber (2012) for variational optimization.

Finally, Combinatorial Multi-Armed Bandits (CMAB), in the stochastic Agarwal and Aggarwal (2018); Rejwan and Mansour (2020) and adversarial Han et al.; Audibert et al. (2014) settings, are somewhat related to our research. Similar to CMAB, we select a subset of size $k$ and receive a subsequent reward. However, in CMAB, it is assumed that each element has a non-constant intrinsic score, and the total reward is a function of the selected elements, which can be linear Audibert et al. (2014); Rejwan and Mansour (2020), monotonically increasing in the element scores Agarwal and Aggarwal (2018), or known in advance Han et al.. The underlying assumption is that selecting elements with better scores is preferable. In our setting, there are no intrinsic element scores, two similar subsets can have vastly different rewards, and the reward for a given subset is constant.

## 2 PRELIMINARIES

Relative weak convexity Zhang and He (2018); Davis and Drusvyatskiy (2019) will be used to obtain the Mirror Descent method's desired guarantees.

**Definition 2.1** (Relative Weak Convexity). Let $f : \mathcal{X} \to \mathbb{R}$. We say that $f$ is $(\rho, \mu)$ relatively weakly convex (RWC) if: (i) $\mu$ is differentiable and 1-strongly convex over $\mathcal{X}$; (ii) The function $f + \rho\mu$ is convex over $\mathcal{X}$.

The following definitions are taken from (Zhang and He, 2018, Section 2.3).

**Definition 2.2** (Bregman Divergence). Let $\mu : \mathcal{X} \to \mathbb{R}$ be differentiable and convex. Then for $\boldsymbol{x}, \boldsymbol{y} \in \mathcal{X}$, the Bregman Divergence is defined as

$$B_\mu(\boldsymbol{x}, \boldsymbol{y}) = \mu(\boldsymbol{x}) - \mu(\boldsymbol{y}) - \langle \nabla\mu(\boldsymbol{y}), \boldsymbol{x} - \boldsymbol{y} \rangle.$$

**Definition 2.3** (Bregman Proximal Operator). Let $f : \mathcal{X} \to \mathbb{R}$ differentiable, and $\mu : \mathcal{X} \to \mathbb{R}$, differentiable and strongly convex, such that $f$ is $(\rho, \mu)$ RWC. Let $\lambda < \rho^{-1}$. Then the Bregman proximal operator is defined as

$$\text{prox}_{\lambda f}(\boldsymbol{z}) := \underset{\boldsymbol{w} \in \mathcal{X}}{\text{argmin}}\, f(\boldsymbol{w}) + \frac{1}{\lambda} B_\mu(\boldsymbol{w}, \boldsymbol{z}).$$

The Bregman Proximal Operator is well-defined and unique for every $z \in \mathcal{X}$.

**Lemma 2.1** ((Zhang and He, 2018, Lemma 2.2)). *Suppose a function $f$ is $(\rho, \mu)$-RWC on $\mathcal{X}$ and $0 < \lambda < \rho^{-1}$. Then for any input $\boldsymbol{z} \in \mathcal{X}$, the function $f + B_\mu(\cdot, \boldsymbol{z})$ is $(\lambda^{-1} - \rho)$-strongly convex. Moreover, the Bregman proximal operator $\text{prox}_{\lambda f}(\boldsymbol{z})$ is unique.*

The Bregman proximal operator is a necessary optimality condition, and is natural when discussing Mirror Descent based algorithms. Since the Bregman proximal operator is unique, the following notion of Bregman Stationarity Measure is well defined.

**Definition 2.4** (Bregman Stationarity Measure). Let $f, \mu, \lambda$ be defined as in Definition 2.3. Then,

$$\Delta_\lambda(\boldsymbol{w}) = \lambda^{-2}(B_\mu(\boldsymbol{w}, \text{prox}_{\lambda f}(\boldsymbol{w})) + B_\mu(\text{prox}_{\lambda f}(\boldsymbol{w}), \boldsymbol{w})).$$

When $\mu$ is Lipschitz continuous, we can derive a connection between the Bregman stationarity measure and the more traditional measure of distance between the subdifferential set and zero (see (Zhang and He, 2018, Equation 2.12)); We provide a detailed proof for completeness.

**Lemma 2.2** (optimality measure). *Let $\hat{\boldsymbol{w}} = \text{prox}_{\lambda f}(\boldsymbol{w})$ and assume that $\mu$ has $M$ Lipschitz gradient. Then, $(\text{dist}\,(0, \partial\,(f + \delta_\mathcal{X})\,(\hat{\boldsymbol{w}})))^2 \leq M \Delta_\lambda(\boldsymbol{w})$, where $\text{dist}\,(\boldsymbol{w}, S) = \underset{\boldsymbol{y} \in S}{\inf} \|\boldsymbol{w} - \boldsymbol{y}\|.$*

## 3  Algorithm

To obtain the desired distribution in our two stochastic formulations ($\text{P}_k$) and ($\text{P}_B$)* we propose a nonconvex Stochastic Mirror Descent (SMD) -based method detailed in Algorithm 1.

For both ($\text{P}_k$) and ($\text{P}_B$) the algorithm we provide: (1) Maintains a parametric distribution of subsets $D_{\boldsymbol{w}}$ ($D_{\boldsymbol{w}} = \psi_{\boldsymbol{w}}$ or $D_{\boldsymbol{w}} = \phi_{\boldsymbol{w}}$); (2) Samples a subset $C \sim D_{\boldsymbol{w}}$; (3) Computes a stochastic gradient estimator $\tilde{\nabla}_{\boldsymbol{w}}^D(C)$ given the subset $C$; and (4) Performs a SMD update using the gradient estimator.

The main challenges in implementing our SMD approach are strongly linked to the choice of the distribution. In particular, the distribution must induce an unbiased stochastic gradient estimator with an explicit formula that can be bounded. Moreover, the expected loss incurred by the distribution must be Relatively Weakly Convex (cf. Section 2). As we shall demonstrate in the sequel, the distributions we propose in Section 4 overcome these challenges.

---

**Algorithm 1:** Stochastic Subset Learner

**Input:** $c > 0$, $\mathcal{X} \subseteq \mathbb{R}^n$, $\boldsymbol{w}_0 \in \mathcal{X}$, $\mu : \mathcal{X} \to \mathbb{R}$, $\{\alpha_t\}_{t=1}^T$

1 **for** $t = 1 \dots, T$ **do**
2     sample $C \sim D_{\boldsymbol{w}_t}$
3     evaluate $\tilde{\nabla}_{\boldsymbol{w}_t}^D(C)$ // `Evaluate the gradient estimator`
4     $\boldsymbol{w}_{t+1} \in \underset{\boldsymbol{w} \in \mathcal{X}}{\text{argmin}}\langle\tilde{\nabla}_{\boldsymbol{w}_t}(C), \boldsymbol{w}\rangle + \frac{1}{\alpha_t} B_\mu(\boldsymbol{w}, \boldsymbol{w}_t)$ // `Mirror Descent step`
5 **end**

6 sample $R \in \{1, \dots, T\}$, such that $\mathbb{P}\,(R = i) = \alpha_i / \sum_{i=t}^T \alpha_t$

7 Return $\boldsymbol{w}_R$

---

*The properties of the formulations and their approximations are discussed in Section 4 and Section 5.

**Remark 3.1** (inputs of Algorithm 1). Applied to our setting and based on the requirements of the SMD, we assume that the inputs of Algorithm 1 obey the following: $\mathcal{X}$ is either $\Delta_n^c$ or $\Delta_{n,k}^c$ and $\mu$ is differentiable and 1-strongly convex. The step size $\alpha_t$ is discussed in Theorem 7.1 and Theorem 7.2.

In Theorem 3.1, we state the convergence guarantees for Algorithm 1 informally and concisely. The full version of this theorem is established in Section 7.

**Theorem 3.1** (Algorithm 1 guarantees (informal)). *Let $x_R$ be the output of Algorithm 1 for one of the supported formulations, with an appropriately chosen step size. Then there exists $\rho > 0$ such that*

$$\mathbb{E}\left[\Delta_{1/2\rho}\left(\boldsymbol{w}_R\right)\right] \leq O\left(\frac{n^{2.5}}{\sqrt{T}}\right),$$

*where the expectation is taken over the random choices of the algorithm.*

We are not aware of any other convergence result in our setting. Even in a CG setting, where the loss function is assumed to be differentiable rather than discrete, we are not aware of any convergence results to stationary points. We note that many applications might find the best sampled subset to be of interest, and indeed, our experiments in Section 8 use this measure.

There are several heuristics which can improve the performance of Algorithm 1. We refer the interested reader to Appendix F.

## 4 Probability Distributions over Subsets

In both formulations, ($P_k$) and ($P_B$), selecting the best subset involves determining a distribution using the decision variable $\boldsymbol{w}$, which we optimize with Algorithm 1. This section presents the proposed probability distributions for these formulations.

For ($P_k$), we define $\psi_{\boldsymbol{w}}$, an unordered weighted-choice without replacement distribution, ensuring subsets of size $k$. To support this, we introduce some notation.

For a subset $C \in \mathcal{C}^k$, let $\pi_1^C, \ldots, \pi_{k!}^C$ denote all permutations of $C$. We simplify to $\pi_1, \ldots, \pi_{k!}$ when $C$ is clear and omit indices when arbitrary. The $l^{\text{th}}$ element of a permutation $\pi$ is $\pi[l]$, or simply $[l]$ when the permutation is clear.

Let $C_1, \ldots, C_N$ ($N = \binom{n}{k}$) be some enumeration of the subsets in $\mathcal{C}^k$. The distribution $\psi_{\boldsymbol{w}}$ is defined by the probability vector $p^{\boldsymbol{w}}$ for a weights vector $\boldsymbol{w} \in \Delta_n^0$ constructed via the probability of choosing the permutation $\pi^{C_i}$ given by

$$p_{i,\pi}^{\boldsymbol{w}} := \mathbb{P}(\pi^{C_i} \mid \boldsymbol{w}) = \prod_{j=1}^{k} \frac{\boldsymbol{w}_{[j]}}{1 - \sum_{l=1}^{j-1} \boldsymbol{w}_{[l]}}. \tag{4.1}$$

The probability of choosing the subset $C_i$ is the sum of the probabilities of choosing its ordered tuples

$$p_i^{\boldsymbol{w}} := \mathbb{P}(C_i \mid \boldsymbol{w}) = \sum_{r=1}^{k!} \mathbb{P}(\pi_r^{C_i} \mid \boldsymbol{w}) = \sum_{r=1}^{k!} p_{i,\pi_r}^{\boldsymbol{w}}. \tag{$\psi_w$}$$

For every choice of $i, k, n, \pi$ and $\boldsymbol{w}$, computing $p_{i,\pi}^{\boldsymbol{w}}$ is possible in $O(k)$ operations. Given that there are $k!$ possible permutations, computing $p_i^{\boldsymbol{w}}$ takes $O(k \cdot k!) = O((k+1)!)$ operations. Therefore, computing this probability is feasible only for *relatively small values* of $k$. On the other hand, note that the probability computations are *completely independent* of $n$.

We now move to define the distribution $\phi_w$ for the problem ($P_B$). Let $C_1, \ldots, C_N$ be some enumeration of the subsets in $\mathcal{C}$. Given a weights vector $\boldsymbol{w} \in \Delta_{n,k}^0$, the distribution $\phi_{\boldsymbol{w}}$ is defined by the probability function

$$\tilde{p}_i^{\boldsymbol{w}} := \mathbb{P}(C_i \mid \boldsymbol{w}) = \prod_{j \in C} \boldsymbol{w}_j \cdot \prod_{j \notin C} (1 - \boldsymbol{w}_j). \tag{$\phi_w$}$$

For every $n, k, i$ and weights vector $\boldsymbol{w}$, calculating $\tilde{p}_i^{\boldsymbol{w}}$ takes $O(n)$ operations.

## 5 RELATIONS BETWEEN THE STOCHASTIC FORMULATIONS AND THEIR APPROXIMATIONS

This section justifies our stochastic formulations by connecting the solution sets of (P), ($P_k$), ($P_B$), and their approximations. These approximations provide bounds on the gradient estimator's moments and the Hessian eigenvalues, essential for the convergence result in Section 7, with proofs in Appendix B.

While it is tempting to use ($P_k$) and ($P_B$) directly, the resulting gradients and gradient estimators can have an arbitrarily large norm. Therefore, we tackle formulations with relaxed upper/lower -bounds on the decision variables instead, that is, we minimize over the sets $\Delta_n^c$ and $\Delta_{n,k}^c$ for $c > 0$. The $c$-approximated problems are:

$$V_k^c := \min_{\boldsymbol{w} \in \Delta_n^c} \mathbb{E}_{C \sim \psi_{\boldsymbol{w}}} \left[ \ell(C) \right], \qquad (P_k^c)$$

$$V_B^c := \min_{\boldsymbol{w} \in \Delta_{n,k}^c} \mathbb{E}_{C \sim \phi_{\boldsymbol{w}}} \left[ \ell(C) \right]. \qquad (P_B^c)$$

Theorem 5.1 provides a bound on the effect of the constraint relaxation in ($P_k^c$) on the optimal value.

**Theorem 5.1** (approximation gap in $k$-cardinality subsets)**.** *Denote* $L_{max} = \max_{C \in \mathcal{C}^k} \ell(C), L^* = \min_{C \in \mathcal{C}^k} \ell(C)$, *and let* $0 < \tau < L_{max} - L^*$ *be some suboptimality gap.*

*The minimum element weight $c$ is defined via an auxiliary variable* $\tilde{c} = c(n - k)$ *and* $\tilde{d} = \prod_{j=1}^{k} \left( 1 - (j - 1) \left( k^{-1} \cdot (1 - \tilde{c}) \right) \right)$. *Define $\tilde{c}^*$ as*

$$\tilde{c}^* = \operatorname*{argmin}_{0 \le \tilde{c} \le 1} \left| \left( 1 - \tilde{d}^{-1} \cdot k! \cdot \left( \frac{1}{k} (1 - \tilde{c}) \right)^k \right) \cdot (L_{max} - L^*) - \tau \right|.$$

*Then for every constraint relaxation of* $0 < c < c^* = \tilde{c}^* \cdot (n - k)^{-1}$, *it holds that* $V_k^c \le L^* + \tau$. *Note that the upper bound of $c$ depends on $\tau$.*

**Theorem 5.2** (approximation gap in $k$-cardinality expected value subsets)**.** *Let* $L_{max} = \max_{C \subseteq \{1,2,\ldots,n\}} \ell(C), L^* = \min_{C \in \mathcal{C}^k} \ell(C) = \inf_{p \in \Delta_N} \mathbb{E}_{i \sim p}[\ell(C_i)]$, *and let* $0 < \tau < L_{max} - L^*$ *be some suboptimality gap. Let $c$ and $1 - c$ be the lower and upper bound of the element weights, respectively. We define $c^*$ as*

$$c^* = (n - k)^{-1} \left( 1 - \sqrt{1 - \tau(L_{max} - L^*)^{-1}} \right).$$

*Then for any constraints relaxation* $0 < c < c^*$ *it holds that* $V_B^c \le L^* + \tau$.

**Remark 5.1.** Note that in both Theorem 5.1 and Theorem 5.2, the upper bound $c^*$ depends on $\tau$. Moreover, $c^*$ is strictly monotonically increasing in $\tau$ for $\tau \in (0, L_{max} - L^*)$.

The elements weight bound $c$ enables us to fine-tune the tradeoff between the suboptimality gap of the approximate problems, and the gradient estimator norm bounds.

## 6 THE UNBIASED STOCHASTIC GRADIENT

We now turn to obtain our gradient estimator. It is derived by analytically calculating the derivative of our proposed distributions. The constructive derivation of the gradient estimator allows us to bound the estimator's size and variance.

We note that the CG framework literature offers several gradient estimators, all of which assume that the loss function is differentiable. To name a few: Sander et al. (2023) introduce an unbiased gradient estimator, Xie and Ermon (2019) present a Gumbel Softmax gradient estimator, and examples of heuristic gradients that are not related to the Gumbel Softmax trick can be found in Ahmed et al. (2022); Pervez et al. (2022). None of the aforementioned gradient estimators provides any theoretical bounds for the size or variance of the gradient estimator.

The stochastic gradient formulas for ($P_k$) and ($P_B$) are defined next, and then proved to be unbiased.

**Definition 6.1** (stochastic gradients)**.** Define the stochastic gradient in the following manner:

If $D_w = \psi_w$. Then for $C_i \sim \psi_{\boldsymbol{w}}$ set

$$\left[\tilde{\nabla}_{\boldsymbol{w}}^{\psi}(C_i)\right]_q = \frac{1}{p_i^{\boldsymbol{w}}} \frac{\partial p_i^{\boldsymbol{w}}}{\partial \boldsymbol{w}_q} \ell(C_i), \qquad \forall q \in \{1, 2, \ldots, n\}. \tag{6.1}$$

The partial derivative is given by

$$\frac{\partial p_i^{\boldsymbol{w}}}{\partial \boldsymbol{w}_q} = \sum_{r=1}^{k!} \frac{\partial p_{i,\pi_r}^{\boldsymbol{w}}}{\partial \boldsymbol{w}_q}, \tag{6.2}$$

where $\pi_1, \ldots, \pi_{k!}$ are all possible permutations of $C_i$ and

$$\frac{\partial p_{i,\pi}^{\boldsymbol{w}}}{\partial \boldsymbol{w}_q} = \begin{cases} \left( (\boldsymbol{w}_{[m]})^{-1} + \sum\limits_{j=m+1}^{k} \left(1 - \sum\limits_{l=1}^{j-1} \boldsymbol{w}_{[l]}\right)^{-1} \right) \cdot p_{i,\pi}^{\boldsymbol{w}}, & q \in C_i, q = \pi^{C_i}[m] =: [m] \\ 0, & \text{otherwise.} \end{cases} \tag{6.3}$$

If $D_w = \phi_w$. Then for $C \sim \phi_{\boldsymbol{w}}$ set

$$\left[\tilde{\nabla}_{\boldsymbol{w}}^{\phi}(C)\right]_j = \begin{cases} \boldsymbol{w}_j^{-1} \ell(C), & j \in C \\ -(1 - \boldsymbol{w}_j)^{-1} \ell(C), & \text{otherwise.} \end{cases} \tag{6.4}$$

The fact that both of the gradient estimators are well-defined and unbiased is established next; it is proved separately for each estimator in Appendix C.

**Lemma 6.1** (gradient estimators properties)**.** *Suppose that $D_w = \psi_w$ or $D_w = \phi_w$. Then the gradient estimators* (6.1) *defined in Definition 6.1 and* (6.4) *defined in Definition 6.1 respectively are well-defined and unbiased, that is,* $\mathbb{E}_{C \sim D_w}\left[\tilde{\nabla}_{\boldsymbol{w}}^{D}(C)\right] = \nabla \mathbb{E}_{C \sim D_w}\left[\ell(C)\right]$, *where* $D \in \{\psi_w, \phi_w\}$.

We conclude this section with an informal remark regarding the way in which the properties of $\ell$ affect the convergence rate of Algorithm 1 empirically.

**Remark 6.1.** Definition 6.1 suggests that the subset loss $\ell(C)$ for any sampled subset $C \in \mathcal{C}$ influences the sampling probabilities of all subsets that share elements with $C$ in subsequent rounds, implicitly assuming that $\ell(C)$ provides insight into the losses of subsets overlapping with $C$. Experimental results in Appendix G indeed indicate that when this assumption holds more strongly, the algorithm performs better.

## 7 CONVERGENCE RESULTS

The SMD convergence analysis using the framework proposed in Zhang and He (2018) suggests that the main challenges in deriving the convergence results lie in establishing the RWC of the objective functions and in obtaining a (stochastic) bound expression for the gradient estimators in Section 6.

The following lemma provides a bound for the gradient and the gradient estimator.

**Lemma 7.1** (gradient estimator bounds)**.** *Let $\boldsymbol{w} \in \Delta_n^0$. Then for every $C \in \mathcal{C}^k$ it holds that:*

1. $\left\|\tilde{\nabla}_{\boldsymbol{w}}^{\psi}(C)\right\|_2 \leq k^{1.5} \cdot |\ell(C)| \cdot \left(\min_j \boldsymbol{w}_j\right)^{-1}$.

2. $\sigma_{max}\left(\nabla^2 \mathbb{E}_{C \sim \psi_{\boldsymbol{w}}}[\ell(C)]\right) \leq n\left(k^2 + k + 1\right) \cdot \max_i |\ell(C_i)| \cdot \left(\min_j \boldsymbol{w}_j\right)^{-2}$.

We now establish bounds utilized when proving the RWC in ($P_B^c$).

**Lemma 7.2.** *Let $\boldsymbol{w} \in \Delta_{n,k}^0$. Then,*

1. $\mathbb{E}_{C \sim \phi_{\boldsymbol{w}}}\left[\left\|\tilde{\nabla}_{\boldsymbol{w}}^{\phi}(C)\right\|_2^2\right] \leq \sum\limits_{j=1}^{n} \left(\frac{1}{\boldsymbol{w}_j} + \frac{1}{1 - \boldsymbol{w}_j}\right) \max_{C \in \mathcal{C}} \ell(C)^2$.

2. *Let* $\rho_B(\boldsymbol{w}) = (n-1) \max_{m,r} \max\{\frac{1}{\boldsymbol{w}_m \boldsymbol{w}_r}, \frac{1}{(1-\boldsymbol{w}_m)(1-\boldsymbol{w}_r)}, \frac{1}{\boldsymbol{w}_m(1-\boldsymbol{w}_r)}\} \max_{C\in\mathcal{C}} |\ell(C)|.$
*Then* $\sigma_{max}\left(\nabla^2 \mathbb{E}_{C \sim \phi_{\boldsymbol{w}}}[\ell(C)]\right) \leq \rho_B(\boldsymbol{w}).$

The RWC of the objectives of the approximations is stated next; it is a corollary of the bounds proved separately for each model in [Appendix D].

**Corollary 7.1** (RWC). *For* $W \in \{\Delta^c_{n,k}, \Delta^c_n\}$ *define* $\mu : W \to \mathbb{R}$ *to be* $\mu(\boldsymbol{w}) = \frac{1}{2}\|\boldsymbol{w}\|^2$, $\mu(\boldsymbol{w}) = -\sum_{i=1}^n \ln(\boldsymbol{w}_i)$, *or* $\mu(\boldsymbol{w}) = \sum_{i=1}^n \boldsymbol{w}_i \ln(\boldsymbol{w}_i)$. *Then* $\mathbb{E}_{C \sim D_{\boldsymbol{w}}}[\ell(C)]$ *is* $(\rho^c_B, \mu)$-*RWC where*

    1. $D = \phi_{\boldsymbol{w}}$, $W = \Delta^c_{n,k}$, $c > 0$ *defined in [Theorem 5.2]*, $\rho^c_B := (n-1)\frac{1}{c^2} \max_{C\in\mathcal{C}} |\ell(C)|$;

    2. $D = \psi_{\boldsymbol{w}}$, $W = \Delta^c_n$, $c > 0$ *defined in [Theorem 5.1]*, $\rho^c_k := c^{-2}n\left(k^2 + k - 1\right) \cdot \max_i |\ell(C_i)|.$

Finally, by utilizing the the gradient estimator's bounds derived above and the RWC property in the problems, we can conclude with the rate guarantees for $(P^c_k)$ and $(P^c_B)$. While the rate depends on both $n$ and $k$, the dependency is *disjoint*, as opposed to the $\binom{n}{k}$. In particular, for small values of $k$, the rate is of order $O\left(n^{2.5}/\sqrt{T}\right)$, as opposed to $O\left(n^k\right)$.

**Theorem 7.1** (($P^c_k$) rate result). *Let* $\boldsymbol{w}_R$ *be the output of [Algorithm 1] for* $(P^c_k)$ *with* $\alpha_t = (n^{2.5}\sqrt{T})^{-1}$ *and let* $\rho = n \cdot c^{-2}\left(k^2 + k - 1\right) \cdot \max_{C\in\mathcal{C}^k} |\ell(C)|$, *where* $c$ *is defined as in* $(P^c_k)$. *Let* $G := \max_{C\in\mathcal{C}^k} \ell(C) - \min_{C\in\mathcal{C}^k} \ell(C)$ *and* $M := \max_{C\in\mathcal{C}^k} |\ell(C)|$. *Then,*

$$\mathbb{E}\left[\Delta_{1/2\rho}(\boldsymbol{w}_R)\right] \leq \left(n^{2.5}G + k^3\left(k^2 + k - 1\right) \cdot M^2 \cdot n^{-1.5} \cdot c^{-4}\right)/\sqrt{T}.$$

*In particular, for* $\tilde{c} = \frac{c^*}{n-k}$ *and* $c^*$ *in [Theorem 5.1]. It holds that*

$$\mathbb{E}\left[\Delta_{1/2\rho}(\boldsymbol{w}_R)\right] \leq O\left(\left(G + k^5\tilde{c}^{-4}M^2\right)n^{2.5}/\sqrt{T}\right),$$

*where* $\tilde{c}$ *does not depend on* $n$.

The rate result for $(P^c_B)$ is independent of $k$, is is therefore suitable for large values of $k$.

**Theorem 7.2** (($P^c_B$) rate result). *Let* $\boldsymbol{w}_R$ *be the output of [Algorithm 1] for* $(P^c_B)$ *with* $\alpha_t = (n^{2.5}\sqrt{T})^{-1}$. *Suppose that* $\rho := n \cdot c^{-2} \max\{|\ell(C)| : C \in \mathcal{C}\}$ *where* $c$ *is defined in [Theorem 5.2]. Let* $\bar{G} := \max_{C\in\mathcal{C}} \ell(C) - \min_{C\in\mathcal{C}} \ell(C)$ *and* $\bar{M} := \max_{C\in\mathcal{C}} |\ell(C)|$. *Then,*

$$\mathbb{E}\left[\Delta_{1/2\rho}(\boldsymbol{w}_R)\right] \leq \frac{2n^{2.5}\bar{G}}{\sqrt{T}} + \frac{2\bar{M}^3}{\sqrt{n}\sqrt{T}\cdot c^3}.$$

*In particular, setting* $\tilde{c} = \frac{c^*}{n-k}$ *where* $c^*$ *is defined in [Theorem 5.2]. It holds that*

$$\mathbb{E}\left[\Delta_{1/2\rho}(\boldsymbol{w}_R)\right] \leq 2\left(\bar{G} + \bar{M}^3\tilde{c}^{-3}\right)n^{2.5}T^{-0.5},$$

*where* $\tilde{c}$ *does not depend on* $n$ *or* $k$.

We conclude this section with a high probability bound for the value of the sampled subset.

**Theorem 7.3.** *Let* $\boldsymbol{w}_R$ *be the distribution parameter returned from [Algorithm 1] applied to either* $(P^c_k)$ *or* $(P^c_B)$, *and let* $D_{\boldsymbol{w}_R} = \psi_{\boldsymbol{w}_R}$ *or* $D_{\boldsymbol{w}_R} = \phi_{\boldsymbol{w}_R}$ *accordingly. Let* $C_1, \ldots, C_m \sim D_{\boldsymbol{w}_R}$ *be independent samples. Denote* $u = \max_{C\in\mathcal{C}} \ell(C)$ *and* $l = \min_{C\in\mathcal{C}} \ell(C)$. *Then for every* $\delta > 0$,

$$\mathbb{P}\left(\min_{i\in\{1,\ldots,m\}} \ell(C_i) - \mathbb{E}_{C\sim D_{\boldsymbol{w}_R}}[\ell(C)] \geq \delta\right) \leq \exp\left\{-\frac{m\delta^2}{2(u-l)^2}\right\}.$$

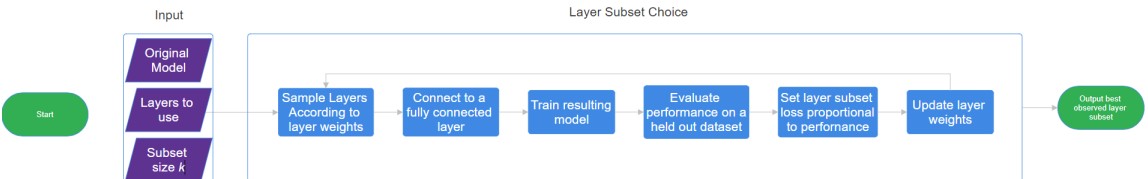

Figure 1: Layer Subset Selection for Transfer Learning Diagram.

## 8 EXPERIMENTS

We conducted experiments in two setups: (i) Transfer Learning (TL) to showcase a practical application, and (ii) Synthetic experiments for direct evaluation on subset selection tasks. TL experiments are discussed below, with synthetic experiment details in Appendix G.

**Experimental Setup.** A natural application of our model is *Transfer Learning* (TL), where knowledge from one task improves performance on a related task Tan et al. (2018); Zhuang et al. (2020). In our setup, the subset selection algorithm selects layers from a frozen pre-trained neural network. At each iteration, a subset of $k$ layers is sampled, connected to a trainable fully connected layer, and the resulting model is trained and evaluated on a held-out dataset. The algorithm computes a loss based on this performance to update selection probabilities. After a preset number of iterations, the process outputs the best subset observed, as shown in Figure 1.

Concretely, we experiment on TL in vision classification tasks based on two models, VisionTransformer (ViT) Dosovitskiy et al. (2020) and ResNet18 He et al. (2016), and two datasets, CIFAR10 Krizhevsky (2009) and SVHN Netzer et al. (2011). The model weights are pre-trained on ImageNet1K Deng et al. (2009). The experiments are restricted to a setting in which the target dataset is small so using a frozen model is preferable to partial or full fine-tunning (see Plested and Gedeon (2022)). Three layer-choosing methods act as benchmarks, in the first two we use hyperparameter tuning (HPO) on the validation set to tune the learning rate and batch size, and in the third we use HPO from the second method. The methods are:

Last1   Using the last layer with learning rate in $[0.001, 0.1]$ and batch size in $\{32, 64, 128\}$.

Last4   Using the last four layers with the same learning rate and batch size ranges.

URand4   Sampling four layers uniformly at random for 50 iterations, training the resulting model on the training set, evaluating on the calibration set, and then selecting the one with the highest accuracy on the calibration set.

We propose the following layer selection methods based on Algorithm 1: For any set of Algorithm 1 hyperparameters ($\tau$ and $\alpha$, where $\alpha_t \equiv \alpha$ for any $t$), we sample layers according to $D_{\boldsymbol{w}}$. For each layer-sample, we construct a model, and train it on the training set with the learning rate and batch size selected by HPO for Last4. The model is then evaluated on the calibration set, and we set the loss in Algorithm 1 to $1 -$ calibration accuracy. We update the layer weights and repeat for $T$ iterations according to Algorithm 1 ($T = 50$). Upon termination of Algorithm 1, the best layers-subset is evaluated on the validation set to update the HPO process for Algorithm 1. Once HPO is finished, we repeat Algorithm 1 for the chosen values of $\tau$, $\alpha$, and evaluate the best in terms of calibration accuracy on the test set. We set $\mu(\boldsymbol{w}) = \|\boldsymbol{w}\|^2$ and use the following to sample layers at each iteration:

CWR4   Sampling four layers with weighted **C**hoice **W**ithout **R**eplacement ($\psi_w$).

HIB4   Sampling layers with **H**eterogeneous **I**ndependent **B**ernoulli experiments ($\phi_w$). Due to memory considerations, samples with more than five layers are assigned accuracy zero.

The performance of each experiment is evaluated for the best set of hyperparameters. Additionally, the three sampling experiments are evaluated on calibration accuracy. For the best set of hyperparameters according to the evaluation accuracy, we show the best observed calibration accuracy per iteration.

The experiments are carried out on AWS Sagemaker, with the instance types "ml.g4dn.16xlarge", with 64 vCPU, 1 Nvidia t4 tensor core GPU, and an Intel Xeon Family physical processor.

|  | ViT-CIFAR10 | | ViT-SVHN | | ResNet18-CIFAR10 | | ResNet18-SVHN | |
|---|---|---|---|---|---|---|---|---|
|  | Mean Acc % | Acc Std | Mean Acc % | Acc Std | Mean Acc % | Acc Std | Mean Acc % | Acc Std |
| Last1 | **91.654** | 0.365 | 50.0369 | 0.971 | 66.698 | 1.134 | 34.700 | 1.345 |
| Last4 | 86.414 | 0.551 | 72.523 | 2.316 | 70.794 | 0.836 | 49.975 | 0.689 |
| URand4 | 89.362 | 1.085 | 75.053 | 2.553 | **74.052** | 1.797 | 53.317 | 12.577 |
| CWR4 | 90.692 | 0.559 | 75.827 | 3.611 | 71.554 | 2.296 | **60.867** | 4.962 |
| HIB4 | 90.724 | 1.045 | **77.781** | 2.036 | 71.336 | 4.102 | 60.478 | 60.478 |

Table 1: Accuracy (Acc) averaged over the 5 subsamples on the original test set. Bold indicates the best.

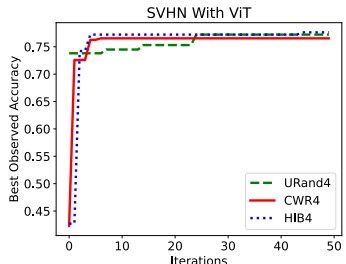 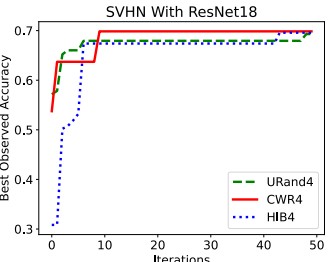

Figure 2: Calibration accuracy per iteration for representative subsamples.

**Experimental Results.** The results in terms of test accuracy are displayed in Table 1. In all cases, selecting the last four layers deterministically (Last4) is inferior to the layer-selection algorithms URand4, CWR4, and HIB4. Among these algorithms, both our methods outperform the random greedy method URand4 in three out of four cases.

The comparison to Last1 is of interest only in the context of TL, as this is the simplest way to use a frozen pre-trained model. In this context, Last1 is the superior method in a single case (ViT with CIFAR10) and inferior to all other methods in all other cases. In the general context of subset selection, the Last1 experiment holds little relevance – our experiments aim to compare methods of selecting subsets of size 4, either deterministically or in expectation.

In a broader sense, our results support the superiority of layer selection algorithms over a deterministic selection in TL on frozen models. This conclusion is compounded by the use of learning rate and batch size selected for Last4, instead of optimizing them directly. We speculate that these results can be extended to more general settings, where the choice of layers can take a different meaning. One example is using a decaying learning rate between layers, as suggested in Plested and Gedeon (2022).

We found that in the calibration accuracy comparisons, one of our methods outperforms the random greedy approach in 14 out of 20 experiments (2 models $\times$ 2 datasets $\times$ 5 subsamples). Actual numbers are omitted as they provide no further insights. The effect of layer selection algorithms is much more pronounced in ResNet18 than in ViT – a representative example for each model is provided in Figure 2. Our experiments also show that CWR4 tends to reach its plateau performance more quickly. We believe this is because its search space is smaller than that of HIB4, and unlike URand4, the initial iterations of CWR4 actively encourage exploration.

## 9 CONCLUSIONS

We introduced a stochastic approach to a non-continuous subset selection formulation facilitating novel theoretical guarantees. Our results motivate both future theoretical and experimental investigations. In particular, further theoretical study of the stochastic model approach, formulation, and possible distributions, as well as other optimization frameworks. Numerical experimentation on TL and other subset selection applications such as Vertex Cover or Independent Set, especially those with challenging non-differentiable metrics, also provide intriguing research prospects.

## 10 ACKNOWLEDGEMENTS

Dan Greenstein's work was partially supported by the ISRAEL SCIENCE FOUNDATION (grant No. 637/21). Nadav Hallak's work was supported by the ISRAEL SCIENCE FOUNDATION (grant No. 637/21).

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

## A PROOFS OF SECTION 2

*Proof.* By the optimality conditions of

$$\text{prox}_{\lambda f}(\boldsymbol{w}) = \underset{z \in \mathcal{X}}{\arg\min}\, f(z) + \frac{1}{\lambda} B_\mu(z, \boldsymbol{w}) = \underset{z \in \mathcal{X}}{\arg\min}\, f(z) + \frac{1}{\lambda}\left(\mu(z) - \mu(\boldsymbol{w}) - \langle \nabla\mu(x), z - \boldsymbol{w} \rangle\right),$$

we have

$$0 \in \nabla f(\hat{\boldsymbol{w}}) + \partial\delta_{\mathcal{X}}(\hat{\boldsymbol{w}}) + \frac{1}{\lambda}\left(\nabla\mu(\hat{\boldsymbol{w}}) - \nabla\mu(\boldsymbol{w})\right) \Rightarrow \frac{1}{\lambda}\left(\nabla\mu(\boldsymbol{w}) - \nabla\mu(\hat{\boldsymbol{w}})\right) \in \partial\left(f + \delta_{\mathcal{X}}\right)(\hat{\boldsymbol{w}}).$$

Hence,

$$\left(\text{dist}\left(0, \partial\left(f + \delta_{\mathcal{X}}\right)(\hat{\boldsymbol{w}})\right)\right)^2 \leq \frac{1}{\lambda^2}\|\nabla\mu(\hat{\boldsymbol{w}}) - \nabla\mu(\boldsymbol{w})\|^2.$$

By (Beck, 2017, Theorem 5.8, part 4), since $\mu$ is assumed to have $M$ Lipschitz gradient over $\mathcal{X}$,

$$\frac{1}{\lambda^2}\|\nabla\mu(\hat{\boldsymbol{w}}) - \nabla\mu(\boldsymbol{w})\|^2 \leq \frac{M}{\lambda^2}\langle\nabla\mu(\boldsymbol{w}) - \nabla\mu(\hat{\boldsymbol{w}}), \boldsymbol{w} - \hat{\boldsymbol{w}}\rangle.$$

Since

$$\Delta_\lambda = \frac{1}{\lambda^2}\langle\nabla\mu(\boldsymbol{w}) - \nabla\mu(\hat{\boldsymbol{w}}), \boldsymbol{w} - \hat{\boldsymbol{w}}\rangle,$$

this concludes the proof. □

## B  PROOFS OF SECTION 5

The following Lemma formally establishes that the minimum of (P) is equal to the infimum of $(\mathrm{P}_k)$, and is lower bounded by the minimum of $(\mathrm{P}_B)$.

**Lemma B.1** (basic relations between formulations). *Denote* $L^* := \inf\limits_{p \in \Delta_N} \mathbb{E}_{i \sim p}[\ell(C_i)] \equiv \sum\limits_{i=1}^{N} p_i \cdot \ell(C_i)$, *and let* $L_B^*$ *be the optimal value of* $(\mathrm{P}_B)$ *and* $L_k^*$ *be the optimal value of* $(\mathrm{P}_k)$. *Suppose that* $C_1, \ldots, C_N$ *is some enumeration of* $\mathcal{C}^k$. *Then:*

1. $L^* = \min\limits_{i=1,\ldots,N} \ell(C_i) \le L_k^*$.

2. *There exists a convergent sequence* $\{w_l\}_{l=1}^{\infty}$ *of feasible solutions to* $(\mathrm{P}_k)$ *such that* $\mathbb{E}_{C \sim \psi_{w_l}}[\ell(C)] \xrightarrow{l \to \infty} L^*$.

3. $L_B^* \le L_k^*$.

*Proof.*  1. Consider the optimization problem

$$\inf_{p \in \Delta_N} \mathbb{E}_{i \sim p}[\ell(C_i)] = \sum_{i=1}^{N} p_i \cdot \ell(C_i).$$

Set

$$p_i^* = \begin{cases} 1, & \text{if } i = i^* \\ 0, & \text{otherwise} \end{cases},$$

where

$$i^* \in \operatorname*{argmin}_{i=1,\ldots,N} \ell(C_i).$$

We note that

$$\sum_{i=1}^{N} p_i \cdot \ell(C_i) \ge \sum_{i=1}^{N} p_i \cdot \min_{i=1,\ldots,N} \ell(C_i) = \min_{i=1,\ldots,N} \ell(C_i),$$

where the second equality follows from $p \in \Delta_N$. Additionally, for $p^*$,

$$\sum_{i=1}^{N} p_i^* \cdot \ell(C_i) = p_{i^*}^* \cdot \ell(C_{i^*}) = \min_{i=1,\ldots,N} \ell(C_i).$$

The relation $L^* \le L_k^*$ follows immediately from the fact that for every $w \in \Delta_n \cap \mathbb{R}_{++}$, $p_i^w \in \Delta_N$, and that the target functions are identical for fixed distributions.

2. Let $i^* \in \operatorname*{argmin}_i \ell(C_i)$. In part 1 of Lemma B.1 it was shown that

$$p^* \in \operatorname*{argmin}_{p \in \Delta_N} \mathbb{E}_{i \sim p}[\ell(C_i)]$$

where $p^*$ is given by

$$p_i^* = \begin{cases} 1, & \text{if } i = i^*, \\ 0, & \text{otherwise.} \end{cases}$$

We construct a convergent sequence $\{w_l\}_{l=1}^{\infty}$, such that

$$p^{w_l} \xrightarrow{l \to \infty} p^*.$$

Since $\mathbb{E}_{i \sim p}[\ell(C_i)]$ is continuous in $p$ and $p^* \in \operatorname*{argmin}_{p \in \Delta_N} \mathbb{E}_{i \sim p}[\ell(C_i)]$, such construction is sufficient for the proof.

Let

$$(\boldsymbol{w}_l)_j = \begin{cases} \dfrac{1}{k} - \dfrac{1}{2k \cdot l}, & \text{if } j \in C_{i^*} \\ \\ \dfrac{1}{2\,(n-k) \cdot l}, & \text{otherwise} \end{cases}.$$

Clearly, for every $l$,

$$\sum_{j=1}^{n} (\boldsymbol{w}_l)_j = \sum_{j \in C_{i^*}} (\boldsymbol{w}_l)_j + \sum_{j \notin C_{i^*}} (\boldsymbol{w}_l)_j$$

$$= k \cdot \left( \frac{1}{k} - \frac{1}{2k \cdot l} \right) + (n-k) \cdot \left( \frac{1}{2\,(n-k) \cdot l} \right) = 1,$$

and for every $l$ and $j$

$$(\boldsymbol{w}_l)_j > 0.$$

Therefore, $\boldsymbol{w}_l \in \Delta_n \cap \mathbb{R}_{++}$. Furthermore, $\boldsymbol{w}_l \xrightarrow{l \to \infty} \boldsymbol{w}^*$, where $\boldsymbol{w}^*$ is given by

$$\boldsymbol{w}_j^* = \begin{cases} \dfrac{1}{k}, & \text{if } j \in C_{i^*} \\ 0, & \text{otherwise} \end{cases}.$$

Next, we calculate $p_{i^*}^{\boldsymbol{w}_l}$

$$p_{i^*}^{\boldsymbol{w}_l} = \sum_{r=1}^{k!} \prod_{j=1}^{k} \frac{1/k - 1/2kl}{1 - (j-1)\,(1/k - 1/2kl)}$$

$$= \sum_{r=1}^{k!} \prod_{j=1}^{k} \frac{2l-1}{2kl - (j-1)\,(2l-1)}$$

$$= \sum_{r=1}^{k!} \prod_{j=1}^{k} \frac{2l-1}{2\,(k-j+1)\,l + j - 1}.$$

Clearly,

$$\frac{2l-1}{2(k-j+1)l + j - 1} \xrightarrow{l \to \infty} \frac{1}{k-j+1}.$$

Therefore it follows that

$$\prod_{j=1}^{k} \frac{2l-1}{2\,(k-j+1)\,l + j - 1} \xrightarrow{l \to \infty} \prod_{j=1}^{k} \frac{1}{k-j+1} = \frac{1}{k!}$$

and

$$\sum_{r=1}^{k!} \prod_{j=1}^{k} \frac{2l-1}{2\,(k-j+1)\,l + j - 1} \xrightarrow{l \to \infty} 1.$$

Since $p_{i^*}^{\boldsymbol{w}_l} \xrightarrow{l \to \infty} 1$ and $p^{\boldsymbol{w}_l} \in \Delta_N$, it follows that for every $i \neq i^*$

$$p_i^{\boldsymbol{w}_l} \xrightarrow{l \to \infty} 0.$$

Hence,

$$p^{\boldsymbol{w}_l} \xrightarrow{l \to \infty} p^*.$$

Since $\mathbb{E}_{i \sim p^{\boldsymbol{w}}} [\ell(C_i)] = \mathbb{E}_{C \sim \psi_{\boldsymbol{w}}} [\ell(C)]$ is continuous in $p^{\boldsymbol{w}}$, it immediately follows that

$$\mathbb{E}_{C \sim \psi_{\boldsymbol{w}_l}} [\ell(C)] \xrightarrow{l \to \infty} \mathbb{E}_{i \sim p^*} [\ell(C_i)] = L^*.$$

3. Let $C_1, \ldots, C_N$ be some enumeration of $\mathcal{C}^k$, and let $i^* \in \operatorname*{argmin}_i \ell(C_i)$.

Define $\bar{\boldsymbol{w}}$ as

$$\bar{\boldsymbol{w}}_j = \begin{cases} 1, & j \in C_{i^*} \\ 0, & \text{otherwise} \end{cases}.$$

Then $\bar{\boldsymbol{w}} \in \{\boldsymbol{w} \in \mathbb{R}_+^n \mid \sum_i^n \boldsymbol{w}_i = k\}$ and

$$\tilde{p}_{i^*}^{\bar{\boldsymbol{w}}} = 1.$$

Therefore, $\bar{\boldsymbol{w}}$ is a feasible solution to ($\mathrm{P}_B$) whose value is $L_k^*$. Hence,

$$L_B^* \le L_k^*.$$

$\square$

*Proof of Theorem 5.1.* Let $C_1, \ldots, C_N$ be some enumeration of $\mathcal{C}^k$, and let $\tilde{i}^* \in \operatorname*{argmin}_i \ell(C_i)$.

Define

$$\bar{\boldsymbol{w}}_j = \begin{cases} \frac{1}{k}\left(1 - c(n-k)\right), & j \in C_{i^*} \\ c, & \text{otherwise} \end{cases},$$

and note that $\bar{\boldsymbol{w}} \in \Delta_n \cap x \ge c$ for $0 \le c \le (n-k)^{-1}$.

The probability of choosing the optimal subset $C_{i^*}$ according to the distribution $D_{\bar{\boldsymbol{w}}}$ is

$$p_{i^*}^{\bar{\boldsymbol{w}}} = \frac{k! \cdot \left(\frac{1}{k}\left(1 - c\left(n-k\right)\right)\right)^k}{\prod_{j=1}^k \left(1 - (j-1)\left(\frac{1}{k}\left(1 - c\left(n-k\right)\right)\right)\right)}.$$

Denoting $\tilde{c} = c\left(n-k\right)$,

$$p_{i^*}^{\bar{\boldsymbol{w}}} = \frac{k! \cdot \left(\frac{1}{k}\left(1 - \tilde{c}\right)\right)^k}{\prod_{j=1}^k \left(1 - (j-1)\left(\frac{1}{k}\left(1 - \tilde{c}\right)\right)\right)}.$$

Note that this is a strictly monotonic decreasing function in $\tilde{c}$ for $0 \le \tilde{c} \le 1$, whose value is $1$ for $\tilde{c} = 0$ and $0$ for $\tilde{c} = 1$. Therefore, $1 - p_{i^*}^{\bar{\boldsymbol{w}}}$ is strictly monotonic increasing in $\tilde{c}$. Since $0 < \tau < L_{\max} - L^*$, it follows that $\frac{\tau}{L_{\max} - L^*} \in (0, 1)$, and therefore there exists a single solution $\tilde{c}^* \in [0, 1]$ to

$$\left(1 - \frac{k! \cdot \left(\frac{1}{k}\left(1 - \tilde{c}\right)\right)^k}{\prod_{j=1}^k \left(1 - (j-1)\left(\frac{1}{k}\left(1 - \tilde{c}\right)\right)\right)}\right) \cdot (L_{max} - L^*) - \tau = 0.$$

Hence, the same $\tilde{c}^*$ minimizes

$$\left|\left(1 - \frac{k! \cdot \left(\frac{1}{k}\left(1 - \tilde{c}\right)\right)^k}{\prod_{j=1}^k \left(1 - (j-1)\left(\frac{1}{k}\left(1 - \tilde{c}\right)\right)\right)}\right) \cdot (L_{max} - L^*) - \tau\right|.$$

We conclude that for every $0 < c \le c^* = \dfrac{\tilde{c}^*}{n - k}$

$$\min_{\boldsymbol{w} \in \Delta_n \cap \boldsymbol{w} \geq c} \mathbb{E}_{C \sim \psi_{\boldsymbol{w}}} \left[ \ell(C) \right] \leq p_{i^*}^{\bar{\boldsymbol{w}}} L^* + (1 - p_{i^*}^{\bar{\boldsymbol{w}}}) L_{max} \leq L^* + \tau.$$

$\square$

*Proof of Theorem 5.2.* Note that $c^*$ is the only root in $[0, (n-k)^{-1}]$ of

$$\left( 1 - (1 - c(n-k))^2 \right) \cdot (L_{max} - L^*) - \tau = 0.$$

Let $C_1, \ldots, C_N$ be some enumeration of subsets of $\mathcal{C}$, and let $i^* \in \underset{i}{\arg\min} \, \ell(C_i)$.

Define $\bar{\boldsymbol{w}}$ as

$$\bar{\boldsymbol{w}}_j = \begin{cases} 1 - \dfrac{c(n-k)}{k}, & j \in C_{i^*} \\ c, & \text{otherwise} \end{cases},$$

and let $\tilde{c} = c \cdot (n-k)$.

We can see that

$$\tilde{p}_{i^*}^{\bar{\boldsymbol{w}}} = \left( 1 - \frac{c(n-k)}{k} \right)^k \cdot (1-c)^{n-k} = \left( 1 - \frac{\tilde{c}}{k} \right)^k \cdot \left( 1 - \frac{\tilde{c}}{n-k} \right)^{n-k}.$$

The sequence $u_m = \left( 1 - \dfrac{\tilde{c}}{m} \right)^m$ is increasing, and therefore for every $k, n$ such that $1 \leq k < n$

$$\left( 1 - \frac{\tilde{c}}{n-k} \right)^{n-k} \geq 1 - \tilde{c}$$

and

$$\left( 1 - \frac{\tilde{c}}{k} \right)^k \geq 1 - \tilde{c},$$

and at least one of the inequalities is strict.

Therefore,

$$\tilde{p}_{i^*}^{\bar{\boldsymbol{w}}} > (1 - \tilde{c})^2 = (1 - c(n-k))^2.$$

Note that $\bar{\boldsymbol{w}} \in \Delta_{n,k}^c$.

It follows that,

$$\min_{\boldsymbol{w} \in \Delta_{n,k}^c} \mathbb{E}_{C \sim \phi_{\boldsymbol{w}}} \left[ \ell(C) \right] \leq (1 - \tilde{c})^2 L^* + \left( 1 - (1 - \tilde{c})^2 \right) L_{max}.$$

For $0 < c \leq c^*$,

$$\min_{\boldsymbol{w} \in \Delta_{n,k}^c} \mathbb{E}_{C \sim \phi_{\boldsymbol{w}}} \left[ \ell(C) \right] < L^* + \tau.$$

$\square$

## C  PROOFS OF SECTION 6

The properties of the gradient estimators for our two formulations summarized in Lemma 6.1 are proved separately in the following two lemmas.

Lemma C.1 establishes that the stochastic gradient (6.1) is well-defined and unbiased.

**Lemma C.1** (gradient estimator for $(\text{P}_k)$). *Let $C_1, \ldots, C_N$ be some enumeration of $\mathcal{C}^k$ and suppose that $D_w = \psi_w$. Then the gradient estimator (6.1) defined in Definition 6.1 is well-defined and unbiased, that is,*

$$\mathbb{E}_{C \sim \psi_{\boldsymbol{w}}} \left[ \tilde{\nabla}_{\boldsymbol{w}}^{\psi} (C) \right] = \nabla \mathbb{E}_{C \sim \psi_{\boldsymbol{w}}} \left[ \ell(C) \right].$$

*Proof.* Let $C_i \in \mathcal{C}^k$. Consider an element index $q \in C_i$ and denote by $\pi$ some permutation of the elements of $C_i$. Since $q \in C_i$, there exists an index $m$, such that $q = \pi[m] \equiv [m]$.

For $j < m$,

$$\frac{\partial}{\partial \boldsymbol{w}_{[m]}} \left( \frac{\boldsymbol{w}_{[j]}}{1 - \sum\limits_{l=1}^{j-1} \boldsymbol{w}_{[l]}} \right) = 0.$$

For $j = m$,

$$\frac{\partial}{\partial \boldsymbol{w}_{[m]}} \left( \frac{\boldsymbol{w}_{[m]}}{1 - \sum\limits_{l=1}^{m-1} \boldsymbol{w}_{[l]}} \right) = \frac{1}{1 - \sum\limits_{l=1}^{m-1} \boldsymbol{w}_{[l]}}.$$

For $j > m$,

$$\frac{\partial}{\partial \boldsymbol{w}_{[m]}} \left( \frac{\boldsymbol{w}_{[j]}}{1 - \sum\limits_{l=1}^{j-1} \boldsymbol{w}_{[l]}} \right) = \frac{\boldsymbol{w}_{[j]}}{\left(1 - \sum\limits_{l=1}^{j-1} \boldsymbol{w}_{[l]}\right)^2}.$$

Recall the derivative rule for product (assuming $f_1(\boldsymbol{w}), \ldots, f_k(\boldsymbol{w}) \neq 0$)

$$\frac{\partial}{\partial \boldsymbol{w}_{[m]}} \left( \prod_{j=1}^{k} f_j(\boldsymbol{w}) \right) = \sum_{t=1}^{k} \frac{\partial f_t(\boldsymbol{w})}{\partial \boldsymbol{w}_{[m]}} \prod_{j=1, j \neq t}^{k} f_j(\boldsymbol{w}) = \sum_{t=1}^{k} \frac{\partial f_t(\boldsymbol{w})}{\partial \boldsymbol{w}_{[m]}} \cdot \frac{1}{f_t(\boldsymbol{w})} \prod_{j=1}^{k} f_j(\boldsymbol{w}).$$

Combining all the above,

$$\frac{\partial p_{i,\pi}^{\boldsymbol{w}}}{\partial \boldsymbol{w}_{[m]}} = \left( \frac{1}{1 - \sum\limits_{l=1}^{m-1} \boldsymbol{w}_{[l]}} \cdot \left( \frac{1 - \sum\limits_{l=1}^{m-1} \boldsymbol{w}_{[l]}}{\boldsymbol{w}_{[m]}} \right) \right.$$

$$\left. + \sum_{j=m+1}^{k} \left( \frac{\boldsymbol{w}_{[j]}}{\left(1 - \sum\limits_{l=1}^{j-1} \boldsymbol{w}_{[l]}\right)^2} \cdot \frac{1 - \sum\limits_{l=1}^{j-1} \boldsymbol{w}_{[l]}}{\boldsymbol{w}_{[j]}} \right) \right) \cdot \prod_{j=1}^{k} \frac{\boldsymbol{w}_{[j]}}{1 - \sum\limits_{l=1}^{j-1} \boldsymbol{w}_{[l]}}.$$

Equivalently,

$$\frac{\partial p_{i,\pi}^{\boldsymbol{w}}}{\partial \boldsymbol{w}_{[m]}} = \left( \frac{1}{\boldsymbol{w}_{[m]}} + \sum_{j=m+1}^{k} \left( \frac{1}{1 - \sum\limits_{l=1}^{j-1} \boldsymbol{w}_{[l]}} \right) \right) \cdot \prod_{j=1}^{k} \frac{\boldsymbol{w}_{[j]}}{1 - \sum\limits_{l=1}^{j-1} \boldsymbol{w}_{[l]}}.$$

Since

$$p_{i,\pi}^{\boldsymbol{w}} = \prod_{j=1}^{k} \frac{\boldsymbol{w}_{[j]}}{1 - \sum\limits_{l=1}^{j-1} \boldsymbol{w}_{[l]}},$$

(6.3) for $q \in C_i$ follows.

On the other hand, if $q \notin C_i$, then $p_{i,\pi}^{\boldsymbol{w}}$ is not a function of $\boldsymbol{w}_q$, and therefore (6.3) for $q \notin C_i$ holds.

The result that the partial derivative $\dfrac{\partial p_i^{\boldsymbol{w}}}{\partial \boldsymbol{w}_q}$ is given by

$$\frac{\partial p_i^{\boldsymbol{w}}}{\partial \boldsymbol{w}_q} = \sum_{r=1}^{k!} \frac{\partial p_{i,\pi_r}^{\boldsymbol{w}}}{\partial \boldsymbol{w}_q}$$

then follows immediately from the fact that $p_i^{\boldsymbol{w}} = \sum_{r=1}^{k!} p_{i,\pi_r}^{\boldsymbol{w}}$, which verifies the correctness of (6.2) and that (6.1) is well-defined.

To prove that (6.1) is unbiased, note that the gradient at index $q$ is given by

$$\nabla \left( \mathbb{E}_{C \sim \psi_{\boldsymbol{w}}} \left[ \ell(C) \right] \right)_q = \frac{\partial}{\partial \boldsymbol{w}_q} \left( \sum_{i=1}^N p_i^{\boldsymbol{w}} \ell(C_i) \right) = \sum_{i=1}^N \frac{\partial p_i^{\boldsymbol{w}}}{\partial \boldsymbol{w}_q} \ell(C_i).$$

On the other hand,

$$\mathbb{E}_{C \sim \psi_{\boldsymbol{w}}} \left[ \tilde{\nabla}_{\boldsymbol{w}}^{\psi} (C) \right]_q = \sum_{i=1}^N p_i^{\boldsymbol{w}} \cdot \left[ \tilde{\nabla}_{\boldsymbol{w}}^{\psi} (C_i) \right]_q = \sum_{i=1}^N p_i^{\boldsymbol{w}} \cdot \frac{1}{p_i^{\boldsymbol{w}}} \frac{\partial p_i^{\boldsymbol{w}}}{\partial \boldsymbol{w}_q} \ell(C_i) = \sum_{i=1}^N \frac{\partial p_i^{\boldsymbol{w}}}{\partial \boldsymbol{w}_q} \ell(C_i).$$

Hence,

$$\mathbb{E}_{C \sim \psi_{\boldsymbol{w}}} \left[ \tilde{\nabla}_{\boldsymbol{w}}^{\psi} (C) \right] = \nabla \left( \mathbb{E}_{C \sim \psi_{\boldsymbol{w}}} \left[ \ell(C) \right] \right).$$

$\square$

The following lemma establishes that the stochastic estimator of the gradient is unbiased for ($P_B$).

**Lemma C.2** (gradient estimator for ($P_B$)). *Suppose that $D_w = \phi_w$. Then the gradient estimator (6.4) defined in Definition 6.1 is unbiased, that is,*

$$\mathbb{E}_{C \sim \phi_{\boldsymbol{w}}} \left[ \tilde{\nabla}_{\boldsymbol{w}}^{\phi} (C) \right] = \nabla \mathbb{E}_{C \sim \phi_{\boldsymbol{w}}} \left[ \ell(C) \right].$$

*Proof of Lemma C.2.* Let $\tilde{C}_1, \ldots, \tilde{C}_{2^n} \in \mathcal{C}$ be an enumeration of all possible subsets of $\{1, \ldots, n\}$.

Let $i$ be the index of the subset $C$. We reformulate $\tilde{\nabla}_{\boldsymbol{w}}^{\phi}(C)$ as

$$\left[ \tilde{\nabla}_{\boldsymbol{w}}^{\phi}(C) \right]_j = \begin{cases} \frac{1}{\tilde{p}_i^{\boldsymbol{w}}} \frac{\ell(C)}{\boldsymbol{w}_j} \tilde{p}_i^{\boldsymbol{w}}, & j \in C \\ \frac{1}{\tilde{p}_i^{\boldsymbol{w}}} \left( -\frac{\ell(C)}{1 - \boldsymbol{w}_j} \right) \tilde{p}_i^{\boldsymbol{w}}, & \text{otherwise.} \end{cases}$$

Since

$$\tilde{p}_i^{\boldsymbol{w}} = \left( \prod_{j \in \tilde{C}_i} \boldsymbol{w}_j \right) \cdot \left( \prod_{k \notin \tilde{C}_i} (1 - \boldsymbol{w}_k) \right),$$

it follows that

$$\frac{\partial \tilde{p}_i^{\boldsymbol{w}}}{\partial \boldsymbol{w}_j} = \begin{cases} \frac{1}{\boldsymbol{w}_j} \tilde{p}_i^{\boldsymbol{w}}, & j \in C, \\ -\frac{1}{1 - \boldsymbol{w}_j} \tilde{p}_i^{\boldsymbol{w}}, & \text{otherwise.} \end{cases}$$

Hence,

$$\left[ \nabla \mathbb{E}_{C \sim \phi_{\boldsymbol{w}}} \left[ \ell(C) \right] \right]_j = \sum_{i=1}^{2^n} \frac{\partial \tilde{p}_i^{\boldsymbol{w}}}{\partial \boldsymbol{w}_j} (C_i) = \sum_{i=1, \, j \in C_i}^{2^n} \frac{1}{\boldsymbol{w}_j} \tilde{p}_i^{\boldsymbol{w}} \ell(C_i) + \sum_{i=1, \, j \notin C_i}^{2^n} -\frac{1}{1 - \boldsymbol{w}_j} \tilde{p}_i^{\boldsymbol{w}} \ell(C_i).$$

On the other hand,

$$\left[ \mathbb{E}_{C \sim \phi_{\boldsymbol{w}}} \left[ \tilde{\nabla}_{\boldsymbol{w}}^{\phi}(C) \right] \right]_j = \sum_{i=1}^{2^n} \tilde{p}_i^{\boldsymbol{w}} \left[ \tilde{\nabla}_{\boldsymbol{w}}^{\phi}(C_i) \right]_j = \sum_{i=1, \, j \in C_i}^{2^n} \frac{1}{\boldsymbol{w}_j} \tilde{p}_i^{\boldsymbol{w}} \ell(C_i) + \sum_{i=1, \, j \notin C_i}^{2^n} -\frac{1}{1 - \boldsymbol{w}_j} \tilde{p}_i^{\boldsymbol{w}} \ell(C_i).$$

Consequently,

$$\mathbb{E}_{C \sim \phi_{\boldsymbol{w}}} \left[ \tilde{\nabla}_{\boldsymbol{w}}^{\phi}(C) \right] = \nabla \mathbb{E}_{C \sim \phi_{\boldsymbol{w}}} \left[ \ell(C) \right].$$

$\square$

# D  PROOFS OF SECTION 7

We start with a technical lemma to bound the component $\dfrac{1}{\boldsymbol{w}_{[m]}} + \sum\limits_{j=m+1}^{k} \left( \dfrac{1}{1 - \sum\limits_{l=1}^{j-1} \boldsymbol{w}_{[l]}} \right)$.

**Lemma D.1.** *Let $\boldsymbol{w} \in \Delta_n \cap \mathbb{R}^n_{++}$, $C \in \mathcal{C}^k$ and let $\pi$ be some permutation on the indices of the elements of $C$. Then,*

$$\frac{1}{\boldsymbol{w}_{[m]}} + \sum_{j=m+1}^{k} \left( \frac{1}{1 - \sum\limits_{l=1}^{j-1} \boldsymbol{w}_{[l]}} \right) \leq \frac{k}{\min\limits_{q \in C} \boldsymbol{w}_q} \leq \frac{k}{\min\limits_{q \in \{1,\ldots,n\}} \boldsymbol{w}_q}.$$

*Proof.* The fact that

$$\frac{1}{\boldsymbol{w}_{[m]}} \leq \frac{1}{\min\limits_{j \in C} \boldsymbol{w}_j} \leq \frac{1}{\min\limits_{q \in \{1,\ldots,n\}} \boldsymbol{w}_q}$$

follows trivially from the definition of the minimum.

Define $w(C) = \sum\limits_{m=1}^{k} \boldsymbol{w}_{[m]}$. Note that $w(C) \leq 1$ since $\boldsymbol{w} \in \Delta_n$. For every permutation $\pi$ of the indices of the elements of $C$ and every $j \leq k$,

$$1 - \sum_{l=1}^{j-1} \boldsymbol{w}_{[l]} \geq w(C) - \sum_{l=1}^{j-1} \boldsymbol{w}_{[l]} \geq \min_{q \in C} \boldsymbol{w}_q \geq \min_{q \in \{1,\ldots,n\}} \boldsymbol{w}_q.$$

Since $\boldsymbol{w} \in \Delta_n \cap \mathbb{R}^n_{++}$, we have

$$\frac{1}{1 - \sum\limits_{l=1}^{j-1} \boldsymbol{w}_{[l]}} \leq \frac{1}{\min\limits_{q \in C} \boldsymbol{w}_q} \leq \frac{1}{\min\limits_{q \in \{1,\ldots,n\}} \boldsymbol{w}_q}.$$

The sum $\sum\limits_{j=m+1}^{k} \left( \dfrac{1}{1 - \sum\limits_{l=1}^{j-1} \boldsymbol{w}_{[l]}} \right)$ has at most $k - 1$ elements. Therefore,

$$\frac{1}{\boldsymbol{w}_{[m]}} + \sum_{j=m+1}^{k} \left( \frac{1}{1 - \sum\limits_{l=1}^{j-1} \boldsymbol{w}_{[l]}} \right) \leq \frac{k}{\min\limits_{q \in C} \boldsymbol{w}_q} \leq \frac{k}{\min\limits_{q \in \{1,\ldots,n\}} \boldsymbol{w}_q}.$$

$\square$

**Lemma D.2.** *Let $\boldsymbol{w} \in \{\boldsymbol{w} \in \mathbb{R}^n \mid 0 < \boldsymbol{w}_j < 1 \ \forall j, \ \sum\limits_{j=1}^{n} \boldsymbol{w}_j = k\}$. Then,*

$$\left\| [\nabla \mathbb{E}_{C \sim \phi_{\boldsymbol{w}}} [\ell(C)]]_j \right\|_\infty \leq \max_{C \in \mathcal{C}} |\ell(C)| \cdot \max_{j \in \{1,\ldots,n\}} \max\{\frac{1}{\boldsymbol{w}_j}, \frac{1}{1 - \boldsymbol{w}_j}\}.$$

*Proof.* By the triangle inequality,

$$
\left| [\nabla \mathbb{E}_{C \sim \phi_{\boldsymbol{w}}} [\ell(C)]]_j \right| = \left| \sum_{i=1,\ j \in C_i}^{2^n} \frac{1}{\boldsymbol{w}_j} \tilde{p}_i^{\boldsymbol{w}} \ell(C_i) + \sum_{i=1,\ j \notin C_i}^{2^n} -\frac{1}{1-\boldsymbol{w}_j} \tilde{p}_i^{\boldsymbol{w}} \ell(C_i) \right|
$$

$$
\leq \sum_{i=1,\ j \in C_i}^{2^n} \left| \frac{1}{\boldsymbol{w}_j} \right| |\tilde{p}_i^{\boldsymbol{w}}| |\ell(C_i)| + \sum_{i=1,\ j \notin C_i}^{2^n} \left| -\frac{1}{1-\boldsymbol{w}_j} \right| |\tilde{p}_i^{\boldsymbol{w}}| |\ell(C_i)|
$$

$$
\leq \max_{C \in \mathcal{C}} |\ell(C)| \sum_{i=1,\ j \in C_i}^{2^n} \left| \frac{1}{\boldsymbol{w}_j} \right| |\tilde{p}_i^{\boldsymbol{w}}| + \sum_{i=1,\ j \notin C_i}^{2^n} \left| -\frac{1}{1-\boldsymbol{w}_j} \right| |\tilde{p}_i^{\boldsymbol{w}}|.
$$

Using the fact that $0 < \boldsymbol{w}_j \leq 1$, we have that $\left| \frac{1}{\boldsymbol{w}_j} \right| = \frac{1}{\boldsymbol{w}_j}$ and $\left| -\frac{1}{1-\boldsymbol{w}_j} \right| = \frac{1}{1-\boldsymbol{w}_j}$. Since $\tilde{p}_i^{\boldsymbol{w}}$ is a probability, $|\tilde{p}_i^{\boldsymbol{w}}| = \tilde{p}_i^{\boldsymbol{w}}$. Therefore,

$$
\left| [\nabla \mathbb{E}_{C \sim \phi_{\boldsymbol{w}}} [\ell(C)]]_j \right| \leq \max_{C \in \mathcal{C}} |\ell(C)| \sum_{i=1,\ j \in C_i}^{2^n} \frac{1}{\boldsymbol{w}_j} \tilde{p}_i^{\boldsymbol{w}} + \sum_{i=1,\ j \notin C_i}^{2^n} \frac{1}{1-\boldsymbol{w}_j} \tilde{p}_i^{\boldsymbol{w}}
$$

$$
\leq \max_{C \in \mathcal{C}} |\ell(C)| \cdot \max\{\frac{1}{\boldsymbol{w}_j}, \frac{1}{1-\boldsymbol{w}_j}\} \cdot \left( \sum_{i=1,\ j \in C_i}^{2^n} \tilde{p}_i^{\boldsymbol{w}} + \sum_{i=1,\ j \notin C_i}^{2^n} \tilde{p}_i^{\boldsymbol{w}} \right).
$$

Therefore, since $\sum_{i=1,\ j \in C_i}^{2^n} \tilde{p}_i^{\boldsymbol{w}} + \sum_{i=1,\ j \notin C_i}^{2^n} \tilde{p}_i^{\boldsymbol{w}} = 1$, we conclude that

$$
\left| [\nabla \mathbb{E}_{C \sim \phi_{\boldsymbol{w}}} [\ell(C)]]_j \right| \leq \max_{C \in \mathcal{C}} |\ell(C)| \cdot \max\{\frac{1}{\boldsymbol{w}_j}, \frac{1}{1-\boldsymbol{w}_j}\}.
$$

Finally, we conclude that

$$
\left\| [\nabla \mathbb{E}_{C \sim \phi_{\boldsymbol{w}}} [\ell(C)]]_j \right\|_\infty \leq \max_{C \in \mathcal{C}} |\ell(C)| \cdot \max_{j \in \{1,\dots,n\}} \max\{\frac{1}{\boldsymbol{w}_j}, \frac{1}{1-\boldsymbol{w}_j}\}.
$$

□

Next, we provide an entrywise bound for the gradient estimator.

**Lemma D.3.** *Let* $\boldsymbol{w} \in \{\boldsymbol{w} \in \mathbb{R}^n \mid 0 < \boldsymbol{w}_j < 1 \ \forall j, \ \sum_{j=1}^n \boldsymbol{w}_j = k\}$. *Then, for every* $C \in \mathcal{C}$,

$$
\left\| \tilde{\nabla}_{\boldsymbol{w}}(C) \right\|_\infty \leq \max_{C \in \mathcal{C}} |\ell(C)| \cdot \max_{j \in \{1,\dots,n\}} \max\{\frac{1}{\boldsymbol{w}_j}, \frac{1}{1-\boldsymbol{w}_j}\}.
$$

*Proof.* By the triangle inequality and the fact that $0 < \boldsymbol{w}_j < 1$, for every $j \in \{1,\dots,n\}$,

$$
\left| \left[ \tilde{\nabla}_{\boldsymbol{w}}(C) \right]_j \right| = \left| \mathbb{1}_{j \in C} \frac{1}{\boldsymbol{w}_j} \ell(C) - \mathbb{1}_{j \notin C} \frac{1}{1-\boldsymbol{w}_j} \ell(C) \right| \leq |\ell(C)| \max\{\frac{1}{\boldsymbol{w}_j}, \frac{1}{1-\boldsymbol{w}_j}\}.
$$

Hence,

$$
\left\| \tilde{\nabla}_{\boldsymbol{w}}(C) \right\|_\infty \leq \max_{C \in \mathcal{C}} |\ell(C)| \cdot \max_{j \in \{1,\dots,n\}} \max\{\frac{1}{\boldsymbol{w}_j}, \frac{1}{1-\boldsymbol{w}_j}\}.
$$

□

An immediate corollary follows from the relation $\|a\|_2 \leq \sqrt{n} \|a\|_\infty$ for every $a \in \mathbb{R}^n$.

**Corollary D.1.** *Let* $\boldsymbol{w} \in \{\boldsymbol{w} \in \mathbb{R}^n \mid 0 < \boldsymbol{w}_j < 1 \ \forall j, \ \sum_{j=1}^{n} \boldsymbol{w}_j = k\}$. *Then, for every* $C \in \mathcal{C}$,

$$\left\| \tilde{\nabla}_{\boldsymbol{w}}(C) \right\|_2 \leq \sqrt{n} \max_{C \in \mathcal{C}} |\ell(C)| \cdot \max_{j \in \{1, \ldots, n\}} \max\{ \frac{1}{\boldsymbol{w}_j}, \frac{1}{1 - \boldsymbol{w}_j} \}.$$

*Proof of Lemma 7.1.* Let $C \in \mathcal{C}^k$. We now prove the two bounds.

1. In order to establish the first bound, we first show that

$$\left\| \tilde{\nabla}_{\boldsymbol{w}}^{\psi}(C) \right\|_{\infty} \leq \frac{k}{\min\limits_{q \in C} \boldsymbol{w}_q} \cdot |\ell(C)|.$$

Let $C_1, \ldots, C_N$ be some enumeration of $\mathcal{C}^k$. Denote the index of $C$ by $i$. Let $q \in \{1, \ldots, n\}$. By the triangle inequality,

$$\left| \left[ \tilde{\nabla}_{\boldsymbol{w}}^{\psi}(C) \right]_q \right| = \left| \frac{1}{p_i^{\boldsymbol{w}}} \sum_{r=1}^{k!} \frac{\partial p_{i, \pi_r}^{\boldsymbol{w}}}{\partial \boldsymbol{w}_q} \ell(C) \right| \leq \left| \frac{1}{p_i^{\boldsymbol{w}}} \right| \left| \sum_{r=1}^{k!} \left| \frac{\partial p_{i, \pi_r}^{\boldsymbol{w}}}{\partial \boldsymbol{w}_q} \right| |\ell(C)| .$$

Since $p_i^{\boldsymbol{w}}$ is the probability of a subset that was sampled, $p_i^{\boldsymbol{w}} > 0$, and therefore $\frac{1}{p_i^{\boldsymbol{w}}} > 0$. By the first part of Definition 6.1, which was proven in Lemma C.1, and by $\boldsymbol{w} \in \Delta_n \cap \mathbb{R}_{++}^n$, it follows that $\frac{\partial p_{i, \pi_r}^{\boldsymbol{w}}}{\partial \boldsymbol{w}_q} \geq 0$. Hence,

$$\left| \frac{1}{p_i^{\boldsymbol{w}}} \right| = \frac{1}{p_i^{\boldsymbol{w}}} \quad \text{and} \quad \left| \frac{\partial p_{i, \pi_r}^{\boldsymbol{w}}}{\partial \boldsymbol{w}_q} \right| = \frac{\partial p_{i, \pi_r}^{\boldsymbol{w}}}{\partial \boldsymbol{w}_q}.$$

Furthermore, if $q \in C_i$ then by Lemma D.1,

$$\frac{\partial p_{i, \pi_r}^{\boldsymbol{w}}}{\partial \boldsymbol{w}_q} \leq \frac{k}{\min\limits_{j} \boldsymbol{w}_j} p_{i, \pi}^{\boldsymbol{w}}.$$

If $q \notin C_i$, then it trivially holds that

$$0 = \frac{\partial p_{i, \pi_r}^{\boldsymbol{w}}}{\partial \boldsymbol{w}_q} \leq \frac{k}{\min\limits_{j} \boldsymbol{w}_j} p_{i, \pi_r}^{\boldsymbol{w}}.$$

Therefore,

$$\frac{\partial p_{i, \pi_r}^{\boldsymbol{w}}}{\partial \boldsymbol{w}_q} \leq \frac{k}{\min\limits_{q \in C} \boldsymbol{w}_q} \cdot p_{i, \pi_r}^{\boldsymbol{w}},$$

and consequently,

$$\left| \left[ \tilde{\nabla}_{\boldsymbol{w}}^{\psi}(C) \right]_q \right| \leq \frac{1}{p_i^{\boldsymbol{w}}} \sum_{r=1}^{k!} \frac{k}{\min\limits_{q \in C} \boldsymbol{w}_q} \cdot p_{i, \pi_r}^{\boldsymbol{w}} |\ell(C)| .$$

Since $\sum_{r=1}^{k!} p_{i, \pi_r}^{\boldsymbol{w}} = p_i^{\boldsymbol{w}}$,

$$\left| \left[ \tilde{\nabla}_{\boldsymbol{w}}^{\psi}(C) \right]_q \right| \leq \frac{k}{\min\limits_{q \in C} \boldsymbol{w}_q} |\ell(C)| ,$$

and we can conclude that

$$\left\| \tilde{\nabla}_{\boldsymbol{w}}^{\psi}(C) \right\|_{\infty} \leq \frac{k}{\min\limits_{q \in C} \boldsymbol{w}_q} \cdot |\ell(C)|. \tag{D.1}$$

By the fact that $\dfrac{\partial p_{i,\pi}^{\boldsymbol{w}}}{\partial \boldsymbol{w}_q} = 0$ for every $q \notin C$, it follows that $\tilde{\nabla}_{\boldsymbol{w}}^{\psi}$ has at most $k$ nonzero

elements. By (D.1), it follows that each entry if bound by $\dfrac{k}{\min\limits_{q \in C} \boldsymbol{w}_q} \cdot |\ell(C)|$. Therefore,

$$\left\| \tilde{\nabla}_{\boldsymbol{w}}^{\psi}(C) \right\|_2 = \sqrt{\sum_{j=1}^{n} \left[ \tilde{\nabla}_{\boldsymbol{w}}^{\psi}(C) \right]_j^2} \le \sqrt{\frac{k^3}{\min\limits_{q \in C} \boldsymbol{w}_q^2} \cdot |\ell(C)|^2} = \frac{k^{1.5}}{\min\limits_{q \in C} \boldsymbol{w}_q} \cdot |\ell(C)|.$$

2. If $q \notin C_i$ or $\tilde{q} \notin C_i$,
$$\frac{\partial^2 p_{i,\pi}^{\boldsymbol{w}}}{\partial \boldsymbol{w}_q \partial \boldsymbol{w}_{\tilde{q}}} = 0.$$

Otherwise, there exist indices $m, o$ such that $q = \pi[m]$, $\tilde{q} = \pi[o]$.

Denote $d_m^{i,\pi}(\boldsymbol{w}) := \sum\limits_{j=m+1}^{k} \left( \dfrac{1}{1 - \sum\limits_{l=1}^{j-1} \boldsymbol{w}_{[l]}} \right)$. Note that

$$\frac{\partial}{\partial \boldsymbol{w}_{[o]}} \left( d_m^{i,\pi}(\boldsymbol{w}) \right) = \sum_{j=\max\{m,o\}+1}^{k} \left( \frac{1}{\left( 1 - \sum\limits_{l=1}^{j-1} \boldsymbol{w}_{[l]} \right)^2} \right). \tag{D.2}$$

If $m = o$, using the derivative rule for multiplication, it follows that

$$\frac{\partial}{\partial \boldsymbol{w}_{[m]}} \left( \frac{\partial p_{i,\pi}^{\boldsymbol{w}}}{\partial \boldsymbol{w}_{[m]}} \right) = \frac{\partial}{\partial \boldsymbol{w}_{[m]}} \left( \left( \frac{1}{\boldsymbol{w}_{[m]}} + d_m^{i,\pi}(\boldsymbol{w}) \right) \cdot p_{i,\pi}^{\boldsymbol{w}} \right) \tag{D.3}$$
$$= \left( \frac{\partial}{\partial \boldsymbol{w}_{[m]}} \left( \frac{1}{\boldsymbol{w}_{[m]}} + d_m^{i,\pi}(\boldsymbol{w}) \right) \right) \cdot p_{i,\pi}^{\boldsymbol{w}}$$
$$+ \left( \frac{1}{\boldsymbol{w}_{[m]}} + d_m^{i,\pi}(\boldsymbol{w}) \right) \cdot \frac{\partial p_{i,\pi}^{\boldsymbol{w}}}{\partial \boldsymbol{w}_{[m]}}.$$

By (6.3),
$$\left( \frac{1}{\boldsymbol{w}_{[m]}} + d_m^{i,\pi}(\boldsymbol{w}) \right) \cdot \frac{\partial p_{i,\pi}^{\boldsymbol{w}}}{\partial \boldsymbol{w}_{[m]}} = \left( \frac{1}{\boldsymbol{w}_{[m]}} + d_m^{i,\pi}(\boldsymbol{w}) \right)^2 p_{i,\pi}^{\boldsymbol{w}}.$$

Plugging into (D.3),

$$\frac{\partial}{\partial \boldsymbol{w}_{[m]}} \left( \frac{\partial p_{i,\pi}^{\boldsymbol{w}}}{\partial \boldsymbol{w}_{[m]}} \right) \tag{D.4}$$
$$= \left( \left( -\frac{1}{\boldsymbol{w}_{[m]}^2} + \frac{\partial d_m^{i,\pi}(\boldsymbol{w})}{\partial \boldsymbol{w}_{[m]}} \right) + \left( \frac{1}{\boldsymbol{w}_{[m]}^2} + \frac{2}{\boldsymbol{w}_{[m]}} \cdot d_m^{i,\pi}(\boldsymbol{w}) + \left( d_m^{i,\pi}(\boldsymbol{w}) \right)^2 \right) \right) \cdot p_{i,\pi}^{\boldsymbol{w}}$$
$$\tag{D.5}$$
$$= \left( \frac{\partial d_m^{i,\pi}(\boldsymbol{w})}{\partial \boldsymbol{w}_{[m]}} + \frac{2}{\boldsymbol{w}_{[m]}} \cdot d_m^{i,\pi}(\boldsymbol{w}) + \left( d_m^{i,\pi}(\boldsymbol{w}) \right)^2 \right) \cdot p_{i,\pi}^{\boldsymbol{w}}.$$

Using the definition of $d_m^{i,\pi}(\boldsymbol{w})$, the fact that $1 - \sum\limits_{l=1}^{j-1} \boldsymbol{w}_{[l]} \ge \min\limits_{j} \boldsymbol{w}_j > 0$ and $\boldsymbol{w}_{[m]} \ge \min\limits_{j} \boldsymbol{w}_j$, and (D.2), it follows that

(a) $\dfrac{\partial d_m^{i,\pi}(\boldsymbol{w})}{\partial \boldsymbol{w}_{[m]}} \le (k-1) \left( \min\limits_{j} \boldsymbol{w}_j \right)^{-2}.$

(b) $\dfrac{2}{\boldsymbol{w}_{[m]}} \cdot d_m^{i,\pi}(\boldsymbol{w}) \leq 2k \left( \min_j \boldsymbol{w}_j \right)^{-2}$.

(c) $\left( d_m^{i,\pi}(\boldsymbol{w}) \right)^2 \leq \left( k^2 - 2k + 1 \right) \left( \min_j \boldsymbol{w}_j \right)^{-2}$.

Combining with (D.4), we get

$$\frac{\partial}{\partial \boldsymbol{w}_{[m]}} \left( \frac{\partial p_{i,\pi}^{\boldsymbol{w}}}{\partial \boldsymbol{w}_{[m]}} \right) \leq \left( k^2 + k \right) \left( \min_j \boldsymbol{w}_j \right)^{-2} p_{i,\pi}^{\boldsymbol{w}}. \tag{D.6}$$

If $o \neq m$, the same arguments lead to

$$\frac{\partial}{\partial \boldsymbol{w}_{[o]}} \left( \frac{\partial p_{i,\pi}^{\boldsymbol{w}}}{\partial \boldsymbol{w}_{[m]}} \right) = \frac{\partial}{\partial \boldsymbol{w}_{[o]}} \left( \left( \frac{1}{\boldsymbol{w}_{[m]}} + d_m^{i,\pi}(\boldsymbol{w}) \right) \cdot p_{i,\pi}^{\boldsymbol{w}} \right) \tag{D.7}$$

$$= \frac{\partial d_m^{i,\pi}(\boldsymbol{w})}{\partial \boldsymbol{w}_{[o]}} p_{i,\pi}^{\boldsymbol{w}} + \left( \frac{1}{\boldsymbol{w}_{[m]}} + d_m^{i,\pi}(\boldsymbol{w}) \right) \left( \frac{1}{\boldsymbol{w}_{[o]}} + d_o^{i,\pi}(\boldsymbol{w}) \right) p_{i,\pi}^{\boldsymbol{w}}.$$

By the definition of $d_m^{i,\pi}(\boldsymbol{w})$, (D.2), $1 - \sum_{l=1}^{j-1} \boldsymbol{w}_{[l]} \geq \min_j \boldsymbol{w}_j > 0$ and $\boldsymbol{w}_{[m]}, \boldsymbol{w}_{[o]} \geq \min_j \boldsymbol{w}_j > 0$,

(a) $\dfrac{\partial d_m^{i,\pi}(\boldsymbol{w})}{\partial \boldsymbol{w}_{[o]}} \leq (k-1) \left( \min_j \boldsymbol{w}_j \right)^{-2}$.

(b) $\dfrac{1}{\boldsymbol{w}_{[m]} \boldsymbol{w}_{[o]}} \leq \left( \min_j \boldsymbol{w}_j \right)^{-2}$.

(c) $\dfrac{1}{\boldsymbol{w}_{[o]}} d_m^{i,\pi}(\boldsymbol{w}) + \dfrac{1}{\boldsymbol{w}_{[m]}} d_o^{i,\pi}(\boldsymbol{w}) \leq 2k \left( \min_j \boldsymbol{w}_j \right)^{-2}$.

(d) $d_m^{i,\pi}(\boldsymbol{w}) d_o^{i,\pi}(\boldsymbol{w}) \leq \left( k^2 - 2k + 1 \right) \left( \min_j \boldsymbol{w}_j \right)^{-2}$.

Plugging into (D.7), it follows that

$$\frac{\partial}{\partial \boldsymbol{w}_{[o]}} \left( \frac{\partial p_{i,\pi}^{\boldsymbol{w}}}{\partial \boldsymbol{w}_{[m]}} \right) \leq \left( k^2 + k + 1 \right) \left( \min_j \boldsymbol{w}_j \right)^{-2} p_{i,\pi}^{\boldsymbol{w}}. \tag{D.8}$$

By (D.4) and (D.7), it also follows that for every $q, \tilde{q} \in C_i$, $\dfrac{\partial^2 p_{i,\pi}^{\boldsymbol{w}}}{\partial q \partial \tilde{q}} > 0$. Since if either $q \notin C_i$ or $\tilde{q} \notin C_i$ it holds that $\dfrac{\partial^2 p_{i,\pi}^{\boldsymbol{w}}}{\partial q \partial \tilde{q}} = 0$, we conclude

$$\frac{\partial^2 p_{i,\pi}^{\boldsymbol{w}}}{\partial q \partial \tilde{q}} \geq 0 \quad \forall q, \tilde{q} \in \{1, \dots, n\}. \tag{D.9}$$

Using the triangle inequality, it immediately follows that

$$\left| \frac{\partial^2}{\partial \boldsymbol{w}_{\tilde{q}} \partial \boldsymbol{w}_q} \left( \mathbb{E}_{C \sim \psi_{\boldsymbol{w}}} \left[ \ell(C) \right] \right) \right| = \left| \sum_{i=1}^{N} \sum_{r=1}^{k!} \frac{\partial^2}{\partial \boldsymbol{w}_{\tilde{q}} \partial \boldsymbol{w}_q} \left( p_{i,\pi_r}^{\boldsymbol{w}} \ell(C_i) \right) \right|$$

$$\leq \sum_{i=1}^{N} \sum_{r=1}^{k!} \left| \frac{\partial^2}{\partial \boldsymbol{w}_{\tilde{q}} \partial \boldsymbol{w}_q} \left( p_{i,\pi_r}^{\boldsymbol{w}} \right) \right| \left| \ell(C_i) \right|$$

$$\leq \sum_{i=1}^{N} \sum_{r=1}^{k!} \frac{\partial^2}{\partial \boldsymbol{w}_{\tilde{q}} \partial \boldsymbol{w}_q} \left( p_{i,\pi_r}^{\boldsymbol{w}} \right) \cdot \max_i \left| \ell(C_i) \right|$$

$$\leq \left( k^2 + k + 1 \right) \cdot \left( \min_j \boldsymbol{w}_j \right)^{-2} \cdot \max_i \left| \ell(C_i) \right|.$$

Let $\rho_k(\boldsymbol{w}) = n\left(k^2+k+1\right) \cdot \left(\min_j \boldsymbol{w}_j\right)^{-2} \cdot \max_i |\ell(C_i)|$. Since

$$\rho_k(\boldsymbol{w}) + \frac{\partial^2}{\partial \boldsymbol{w}_i \partial \boldsymbol{w}_i}\left(\mathbb{E}_{C \sim \psi_{\boldsymbol{w}}}\left[\ell(C)\right]\right) \geq \rho_k(\boldsymbol{w}) - \left(k^2+k+1\right) \cdot \left(\min_j \boldsymbol{w}_j\right)^{-2} \cdot \max_i |\ell(C_i)|,$$

and

$$\rho_k(\boldsymbol{w}) - \frac{\partial^2}{\partial \boldsymbol{w}_i \partial \boldsymbol{w}_i}\left(\mathbb{E}_{C \sim \psi_{\boldsymbol{w}}}\left[\ell(C)\right]\right) \geq \rho_k(\boldsymbol{w}) - \left(k^2+k+1\right) \cdot \left(\min_j \boldsymbol{w}_j\right)^{-2} \cdot \max_i |\ell(C_i)|,$$

and in addition

$$\max_i \sum_{j=1,\ j \neq i}^{n} \left|\left[\nabla^2 \mathbb{E}_{C \sim \psi_{\boldsymbol{w}}}\left[\ell(C)\right]\right]_{i,j}\right| \leq (n-1)\left(k^2+k+1\right) \cdot \left(\min_j \boldsymbol{w}_j\right)^{-2} \cdot \max_i |\ell(C_i)|,$$

it follows that for all $i$

$$\rho_k(\boldsymbol{w}) + \left[\nabla^2 \mathbb{E}_{C \sim \psi_{\boldsymbol{w}}}\left[\ell(C)\right]\right]_{i,i} \geq \sum_{j=1,\ j \neq i}^{n} \left|\left[\nabla^2 \mathbb{E}_{C \sim \psi_{\boldsymbol{w}}}\left[\ell(C)\right]\right]_{i,j}\right|,$$

and

$$\rho_k(\boldsymbol{w}) - \left[\nabla^2 \mathbb{E}_{C \sim \psi_{\boldsymbol{w}}}\left[\ell(C)\right]\right]_{i,i} \geq \sum_{j=1,\ j \neq i}^{n} \left|\left[\nabla^2 \mathbb{E}_{C \sim \psi_{\boldsymbol{w}}}\left[\ell(C)\right]\right]_{i,j}\right|.$$

Therefore, both $\rho_k(\boldsymbol{w})I + \nabla^2 \mathbb{E}_{C \sim \psi_{\boldsymbol{w}}}\left[\ell(C)\right]$ and $\rho_k(\boldsymbol{w})I - \mathbb{E}_{C \sim \psi_{\boldsymbol{w}}}\left[\ell(C)\right]$ are diagonally dominant. Since diagonally dominant matrices are positive definite, it follows that

$$\rho_k(\boldsymbol{w})I + \nabla^2 \mathbb{E}_{C \sim \psi_{\boldsymbol{w}}}\left[\ell(C)\right] \succ 0 \Rightarrow \nabla^2 \mathbb{E}_{C \sim \psi_{\boldsymbol{w}}}\left[\ell(C)\right] \succ -\rho_k(\boldsymbol{w})I,$$

$$\rho_k(\boldsymbol{w})I - \nabla^2 \mathbb{E}_{C \sim \psi_{\boldsymbol{w}}}\left[\ell(C)\right] \succ 0 \Rightarrow \rho_k(\boldsymbol{w})I \succ \nabla^2 \mathbb{E}_{C \sim \psi_{\boldsymbol{w}}}\left[\ell(C)\right].$$

In conclusion,

$$\rho_k(\boldsymbol{w})I \succ \nabla^2 \mathbb{E}_{C \sim \psi_{\boldsymbol{w}}}\left[\ell(C)\right] \succ -\rho_k(\boldsymbol{w})I.$$

$\square$

*Proof of Lemma 7.2.* We prove the two parts.

1. For every $j \in \{1, \ldots, n\}$ by the law of total expectation and the definition of $\tilde{\nabla}_{\boldsymbol{w}}^{\phi}(C)$,

$$\mathbb{E}_{C \sim \phi_{\boldsymbol{w}}}\left[\left(\left[\tilde{\nabla}_{\boldsymbol{w}}^{\phi}(C)\right]_j\right)^2\right]$$

$$= \mathbb{P}\left(\text{j is chosen}\right) \cdot \frac{1}{\boldsymbol{w}_j^2} \cdot \mathbb{E}_{C \sim \phi_{\boldsymbol{w}} \mid \text{j is chosen}}\left[\ell(C)^2\right]$$

$$+ \left(1 - \mathbb{P}\left(\text{j is chosen}\right)\right) \cdot \frac{1}{\left(1 - \boldsymbol{w}_j\right)^2} \cdot \mathbb{E}_{C \sim \phi_{\boldsymbol{w}} \mid \text{j is not chosen}}\left[\ell(C)^2\right].$$

Since

$$\mathbb{P}\left(\text{j is chosen}\right) = \boldsymbol{w}_j$$

and

$$\max\{\mathbb{E}_{C \sim \phi_{\boldsymbol{w}} \mid \text{j is chosen}}\left[\ell(C)^2\right], \mathbb{E}_{C \sim \phi_{\boldsymbol{w}} \mid \text{j is not chosen}}\left[\ell(C)^2\right]\} \leq \max_{C \in \mathcal{C}} \ell(C)^2,$$

we can derive that

$$\mathbb{E}_{C \sim \phi_{\boldsymbol{w}}}\left[\left(\left[\tilde{\nabla}_{\boldsymbol{w}}^{\phi}(C)\right]_j\right)^2\right] \leq \left(\frac{1}{\boldsymbol{w}_j} + \frac{1}{1 - \boldsymbol{w}_j}\right) \max_{C \in \mathcal{C}} \ell(C)^2.$$

Therefore,

$$\mathbb{E}_{C \sim \phi_{\boldsymbol{w}}}\left[\left\|\tilde{\nabla}_{\boldsymbol{w}}^{\phi}(C)\right\|_2^2\right] \leq \sum_{j=1}^{n}\left(\frac{1}{\boldsymbol{w}_j} + \frac{1}{1 - \boldsymbol{w}_j}\right) \max_{C \in \mathcal{C}} \ell(C)^2.$$

2. Recall that

$$\mathbb{E}_{C \sim \phi_{\boldsymbol{w}}}[\ell(C)] = \sum_{i=1}^{2^n} \prod_{j \in C_i} \boldsymbol{w}_j \prod_{j \notin C_i} (1 - \boldsymbol{w}_j),$$

and that

$$\tilde{p}_i^{\boldsymbol{w}} = \prod_{j \in C_i} \boldsymbol{w}_j \prod_{j \notin C_i} (1 - \boldsymbol{w}_j).$$

For every $i$ and every $j$,

$$\frac{\partial^2 \tilde{p}_i^{\boldsymbol{w}}}{\partial \boldsymbol{w}_j^2} = 0,$$

and hence for every $j \in \{1, \dots, n\}$

$$\nabla^2 \left[ \mathbb{E}_{C \sim \phi_{\boldsymbol{w}}}[\ell(C)] \right]_{jj} = 0.$$

For $j \neq l$, if $j, l \in C_i$,

$$\frac{\partial^2 \tilde{p}_i^{\boldsymbol{w}}}{\partial \boldsymbol{w}_j \partial \boldsymbol{w}_l} = \frac{1}{\boldsymbol{w}_j \boldsymbol{w}_l} \tilde{p}_i^{\boldsymbol{w}}.$$

If $j \in C_i, l \notin C_i$,

$$\frac{\partial^2 \tilde{p}_i^{\boldsymbol{w}}}{\partial \boldsymbol{w}_j \partial \boldsymbol{w}_l} = -\frac{1}{\boldsymbol{w}_j (1 - \boldsymbol{w}_l)} \tilde{p}_i^{\boldsymbol{w}}.$$

If $j \notin C_i, l \in C_i$,

$$\frac{\partial^2 \tilde{p}_i^{\boldsymbol{w}}}{\partial \boldsymbol{w}_j \partial \boldsymbol{w}_l} = -\frac{1}{(1 - \boldsymbol{w}_j) \boldsymbol{w}_l} \tilde{p}_i^{\boldsymbol{w}}.$$

If $j, l \notin C_i$,

$$\frac{\partial^2 \tilde{p}_i^{\boldsymbol{w}}}{\partial \boldsymbol{w}_j \partial \boldsymbol{w}_l} = \frac{1}{(1 - \boldsymbol{w}_j)(1 - \boldsymbol{w}_l)} \tilde{p}_i^{\boldsymbol{w}}.$$

Therefore,

$$\frac{\partial^2}{\partial \boldsymbol{w}_j \partial \boldsymbol{w}_l} \mathbb{E}_{C \sim \phi_{\boldsymbol{w}}}[\ell(C)]$$

$$= \sum_{i=1}^{2^n} \left( 1_{j,l \in C_i} \cdot \frac{1}{\boldsymbol{w}_j \boldsymbol{w}_l} - 1_{j \in C_i, l \notin C_i} \frac{1}{\boldsymbol{w}_j (1 - \boldsymbol{w}_l)} - 1_{j \notin C_i, l \in C_i} \frac{1}{(1 - \boldsymbol{w}_j) \boldsymbol{w}_l} + 1_{j,l \notin C_i} \frac{1}{(1 - \boldsymbol{w}_j)(1 - \boldsymbol{w}_l)} \right) \tilde{p}_i^{\boldsymbol{w}} \ell(C_i).$$

Using the facts that

$$-\max_{m,r} \frac{1}{\boldsymbol{w}_m (1 - \boldsymbol{w}_r)} \leq -\frac{1}{\boldsymbol{w}_j (1 - \boldsymbol{w}_l)} < 0$$

and

$$0 \leq \max\{\frac{1}{\boldsymbol{w}_j \boldsymbol{w}_l}, \frac{1}{(1 - \boldsymbol{w}_j)(1 - \boldsymbol{w}_l)}\} \leq \max_{m,r} \max\{\frac{1}{\boldsymbol{w}_m \boldsymbol{w}_r}, \frac{1}{(1 - \boldsymbol{w}_m)(1 - \boldsymbol{w}_r)}\},$$

as well as the fact that for every $i$

$$\ell(C_i) \leq \max_{C \in \mathcal{C}} |\ell(C)|$$

and that

$$\sum_{i=1}^{2^n} \tilde{p}_i^{\boldsymbol{w}} = 1,$$

we can deduce that

$$-\max_{m,r} \frac{1}{\boldsymbol{w}_m (1 - \boldsymbol{w}_r)} \cdot \max_{C \in \mathcal{C}} |\ell(C)| \leq \frac{\partial^2}{\partial \boldsymbol{w}_j \partial \boldsymbol{w}_l} \mathbb{E}_{C \sim \phi_{\boldsymbol{w}}}[\ell(C)]$$

$$\leq \max_{m,r} \max\{\frac{1}{\boldsymbol{w}_m \boldsymbol{w}_r}, \frac{1}{(1 - \boldsymbol{w}_m)(1 - \boldsymbol{w}_r)}\} \max_{C \in \mathcal{C}} |\ell(C)|.$$

Consequently,

$$\left| \frac{\partial^2}{\partial \boldsymbol{w}_j \partial \boldsymbol{w}_l} \mathbb{E}_{C \sim \phi_{\boldsymbol{w}}} \left[ \ell(C) \right] \right| \tag{D.10}$$

$$\leq \max_{m,r} \max \{ \frac{1}{\boldsymbol{w}_m \boldsymbol{w}_r}, \frac{1}{(1-\boldsymbol{w}_m)(1-\boldsymbol{w}_r)}, \frac{1}{\boldsymbol{w}_m (1-\boldsymbol{w}_r)} \} \max_{C \in \mathcal{C}} |\ell(C)| \, .$$

By (D.10), for every $j$,

$$\rho_B(\boldsymbol{w}) \geq \sum_{l=1, l \neq j}^{n} \left| \frac{\partial^2}{\partial \boldsymbol{w}_j \partial \boldsymbol{w}_l} \mathbb{E}_{C \sim \phi_{\boldsymbol{w}}} \left[ \ell(C) \right] \right|,$$

and hence $\nabla^2 \mathbb{E}_{C \sim \phi_{\boldsymbol{w}}} \left[ \ell(C) \right] + \rho_B(\boldsymbol{w}) I$ is diagonally dominant. Since every diagonally dominant matrix is positive semi-definite, it follows that

$$\nabla^2 \mathbb{E}_{C \sim \phi_{\boldsymbol{w}}} \left[ \ell(C) \right] + \rho_B(\boldsymbol{w}) I \succeq 0$$

and therefore

$$\lambda_{min} \left( \nabla^2 \mathbb{E}_{C \sim \phi_{\boldsymbol{w}}} \left[ \ell(C) \right] \right) \geq -\rho_B(\boldsymbol{w}).$$

For similar reasons, $\rho_B(\boldsymbol{w}) - \nabla^2 \mathbb{E}_{C \sim \phi_{\boldsymbol{w}}} \left[ \ell(C) \right]$ is also diagonally dominant, and hence

$$\rho_B(\boldsymbol{w}) I - \nabla^2 \mathbb{E}_{C \sim \phi_{\boldsymbol{w}}} \left[ \ell(C) \right] \succeq 0,$$

which leads to

$$\lambda_{max} \left( \nabla^2 \mathbb{E}_{C \sim \phi_{\boldsymbol{w}}} \left[ \ell(C) \right] \right) \leq \rho_B(\boldsymbol{w}).$$

$\square$

*Proof of Theorem 7.1.* The result follows immediately from (Zhang and He, 2018, Corollary 3.1), with the parameters $\rho = \dfrac{n \left( k^2 + k - 1 \right)}{c^2} \cdot \max_{C \in \mathcal{C}^k} |\ell(C)|$ (as proven in Corollary 7.1), $L = \dfrac{k^{1.5}}{c} \max_{C \in \mathcal{C}^k} |\ell(C)|$ (proven in Lemma 7.1), $T_{1/2\rho}(\boldsymbol{w}_0) \leq \max_{C \in \mathcal{C}^k} \ell(C), T_{min} \geq \min_{C \in \mathcal{C}^k} \ell(C).$

Using the fact that $c \leq \dfrac{\tilde{c}}{n-k}$, where $\tilde{c}$ is independent of $n$, yields that particular result. $\square$

*Proof of Theorem 7.2.* The result follows immediately from (Zhang and He, 2018, Corollary 3.1), with the parameters $\rho = n \dfrac{1}{c^2} \max_{C \in \mathcal{C}} |\ell(C)|$ (as proven in Corollary 7.1), $L^2 = \dfrac{2n}{c} \max_{C \in \mathcal{C}} \ell(C)^2$ (proven in Lemma 7.2), $T_{1/2\rho}(\boldsymbol{w}_0) \leq \max_{C \in \mathcal{C}^k} \ell(C), T_{min} \geq \min_{C \in \mathcal{C}^k} \ell(C).$ $\square$

*Proof of Theorem 7.3.* Note that

$$\min_{i \in \{1,\dots,m\}} \ell(C_i) - \mathbb{E}_{C \sim D_{\boldsymbol{w}_R}} \left[ \ell(C) \right] \geq \delta \Leftrightarrow \ell(C_i) - \mathbb{E}_{C \sim D_{\boldsymbol{w}_R}} \left[ \ell(C) \right] \geq \delta \quad \forall i.$$

By Hoeffding's inequality, for all $i$,

$$\mathbb{P}\left( \ell(C_i) - \mathbb{E}_{C \sim D_{\boldsymbol{w}_R}} \left[ \ell(C) \right] \geq \delta \right) \leq \exp \left\{ -\frac{\delta^2}{2(u-l)} \right\}.$$

Since $C_1, \dots, C_m$ are i.i.d,

$$\mathbb{P}\left( \min_{i \in \{1,\dots,m\}} \ell(C_i) - \mathbb{E}_{C \sim D_{\boldsymbol{w}_R}} \left[ \ell(C) \right] \geq \delta \right) = \mathbb{P}\left( \ell(C_i) - \mathbb{E}_{C \sim D_{\boldsymbol{w}_R}} \left[ \ell(C) \right] \geq \delta \right)^m$$

$$\leq \exp \left\{ -\frac{m\delta^2}{2(u-l)} \right\}.$$

$\square$

# E    MIRROR DESCENT STEP CALCULATION ALGORITHMS

Let $0 < c < \dfrac{1}{n}$. We consider the mirror descent step for all the combinations of

1. $\mathcal{X}_1 := \Delta_n^c = \{x \in \Delta_n \mid c \leq x_j \ \forall j \in \{1, \dots, n\}\}$

2. $\mathcal{X}_2 := \Delta_{n,k}^c = \{x \in \mathbb{R}^n \mid c \leq x_j \leq 1 - c \ \forall j \in \{1, \dots, n\}, \ \sum\limits_{j=1}^{n} x_j = k\}$

and $\mu_i : \mathcal{X}_j \to \mathbb{R}$ for $i \in \{1, 2, 3\}$ and $j \in \{1, 2\}$, where

1. $\mu_1(x) = \dfrac{1}{2} \|x\|^2.$

2. $\mu_2(x) = - \sum\limits_{i=1}^{n} \ln(x_i).$

3. $\mu_3(x) = \sum\limits_{i=1}^{n} x_i \ln(x_i).$

**Lemma E.1.** *Let $\mu_1 : \Delta_n^c \to \mathbb{R}$ be defined as $\mu_1(x) = \dfrac{1}{2} \|x\|^2$. Then,*

$$\operatorname*{argmin}_{z \in \Delta_n^c} \langle g, z \rangle + \frac{1}{\alpha} B_{\mu_1}(x, z) = \{z^*\},$$

*where $z^*$ is given by*

$$z_i^* = \begin{cases} x_i - \alpha \left(g_i + \mu^*\right), & \mu^* \leq \dfrac{x_i - c}{\alpha} - g_i \\ c, & otherwise \end{cases},$$

*and $\mu^*$ is the unique solution of*

$$\sum_{i:\ \mu > (x_i - c)/\alpha - g_i} c + \sum_{i:\ \mu \leq (x_i - c)/\alpha - g_i} (x_i - \alpha(\mu + g_i)) = 1.$$

*Furthermore, $\mu^*$ can be found using bisection.*

*Proof.* We start by proving that $\mu^*$ exists, is unique, and can be found using bisection. The function is continuous, strictly decreasing when there exists at least one index $i$ such that $\mu \leq (1 - c)/\alpha - g_i$, and

$$\sum_{i:\ \mu > (x_i - c)/\alpha - g_i} c + \sum_{i:\ \mu \leq (x_i - c)/\alpha - g_i} (x_i - \alpha(\mu + g_i)) \xrightarrow{\mu \to -\infty} \infty,$$

$$\sum_{i:\ \mu > (x_i - c)/\alpha - g_i} c + \sum_{i:\ \mu \leq (x_i - c)/\alpha - g_i} (x_i - \alpha(\mu + g_i)) \xrightarrow{\mu \to \infty} = n \cdot c - 1 < 0.$$

Therefore, $\mu^*$ exists, is unique, and can be found using bisection.

Since $\langle g, x \rangle + B_{\mu_1}(x, z)$ is continuous and $\Delta_n^c$ is compact, a minimizer exists. Since $\langle g, x \rangle + B_{\mu_1}(x, z)$ is strongly convex, the minimizer is unique. Furthermore, due to the convexity, the KKT conditions are sufficient.

Define the Lagrangian $L : \mathbb{R}^n \times \mathbb{R}_+^n \times \mathbb{R}$,

$$L(z, \lambda, \mu) = \langle g, z \rangle + \frac{1}{2\alpha} \|z - x\|^2 + \mu \left( \sum_{j=1}^{n} z_j - 1 \right) + \sum_{j=1}^{n} \lambda_j (c - z_j).$$

The KKT conditions are

1. $g_j + \dfrac{1}{\alpha} z_j - \dfrac{1}{\alpha} x_j + \mu - \lambda_j = 0$ for all $j \in \{1, \dots, n\}$.

2. $\lambda_j(c - z_j) = 0$ for all $j \in \{1, \ldots, n\}$.

3. $\sum_{j=1}^{n} z_j = 1$.

4. $z_j \geq c$ for all $j \in \{1, \ldots, n\}$.

5. $\lambda \geq 0$.

We can see that $z^*, \mu^*, \lambda^*$, where $\lambda^*$ is given by

$$\lambda_j^* = \begin{cases} 0, & \mu^* \leq \dfrac{x_j - c}{\alpha} - g_j \\ \mu^* - \dfrac{x_j - c}{\alpha} + g_j, & \text{otherwise} \end{cases},$$

is a valid solution to the KKT system. Hence, $z^*$ is the optimal solution. $\qquad\square$

**Lemma E.2.** *Let* $\mu_2 : \Delta_n^c \to \mathbb{R}$ *be defined as* $\mu_2(x) = -\sum_{i=1}^{n} \ln(x_i)$. *Assume that* $x \in \Delta_n^c$. *Then,*

$$\operatorname*{argmin}_{z \in \Delta_n^c} \langle g, z \rangle + \frac{1}{\alpha} B_{\mu_2}(x, z) = \{z^*\},$$

*where* $z^*$ *is given by*

$$z_i^* = \max\{\frac{1}{1/x_i + \alpha\,(g_i + \mu^*)}, c\}$$

*and* $\mu^*$ *is the unique solution of*

$$\sum_{i=1}^{n} \max\{\frac{1}{1/x_i + \alpha\,(g_i + \mu)}, c\} = 1$$

*on* $\mu \in (\max_j -\dfrac{1}{\alpha x_j} - g_j, \infty)$. *Furthermore,* $\mu^*$ *can be found using bisection.*

*Proof.* We start by proving that $\mu^*$ exists, is unique, and can be found using bisection. The function is continuous. Additionally,

$$\sum_{i=1}^{n} \max\{\frac{1}{1/x_i + \alpha\,(g_i + \mu^*)} \xrightarrow{\mu \to \infty} = n \cdot c < 1,$$

and

$$\sum_{i=1}^{n} \max\{\frac{1}{1/x_i + \alpha\,(g_i + \mu^*)} \xrightarrow{\mu \to (\max_j -\frac{1}{\alpha x_j} - g_j)^+} = \infty,$$

Therefore, a solution $\mu^*$ exists.

For $\mu > \max_j -\dfrac{1}{\alpha x_j} - g_j$, $\dfrac{1}{1/x_i + \alpha\,(g_i + \mu^*)}$ is strictly decreasing.

Therefore, the solution $\mu^*$ is unique on $(\max_j -\dfrac{1}{\alpha x_j} - g_j, \infty)$, and can be found using bisection.

Define the Lagrangian $L : \mathbb{R}^n \times \mathbb{R}_+^n \times \mathbb{R}$,

$$L(z, \lambda, \mu) = \sum_{i=1}^{n} \left( \left(g_i + \frac{1}{\alpha x_i} + \mu - \lambda_i\right) z_i - \frac{1}{\alpha} \ln(z_i) \right) - \mu + c \sum_{i=1}^{n} \lambda_i.$$

The problem is continuous and the feasible set is compact, therefore an optimal solution exists. Since the problem is also convex, the KKT conditions are sufficient for optimality. The KKT conditions are given by

1. $g_j + \dfrac{1}{\alpha x_j} + \mu - \lambda_j - \dfrac{1}{\alpha}\dfrac{1}{z_j} = 0$ for all $j \in \{1, \ldots, n\}$.

2. $\lambda_j(c - z_j) = 0$ for all $j \in \{1, \ldots, n\}$.

3. $\sum\limits_{j=1}^{n} z_j = 1$.

4. $z_j \geq c$ for all $j \in \{1, \ldots, n\}$.

5. $\lambda \geq 0$.

A solution to the KKT conditions is given by $(z^*, \lambda^*, \mu^*)$, where $z^*$ and $\mu^*$ were defined above and

$$
\lambda_j^* = \begin{cases} 0, & z_j^* > c \\ g_j + \mu^* + \dfrac{1}{\alpha x_j} - \dfrac{1}{\alpha c}, & \text{otherwise} \end{cases}.
$$

To complete the proof, we will prove that $\lambda^* \geq 0$. If $z_j^* > c$, then $\lambda_j^* \geq 0$ trivially. If $z_j^* = c$, then

$$
\frac{1}{1/x_j + \alpha\left(g_j + \mu^*\right)} \leq c. \tag{E.1}
$$

Since both sides of the inequality are non-negative, it follows that

$$
\frac{1}{\alpha c} \leq \frac{1}{\alpha x_j} + g_j + \mu^*,
$$

and therefore

$$
\lambda_j^* = g_j + \mu^* + \frac{1}{\alpha x_j} - \frac{1}{\alpha c} \geq 0.
$$

$\square$

**Lemma E.3.** *Let $\mu_3 : \Delta_n^c \to \mathbb{R}$ be defined as $\mu_2(x) = \sum\limits_{i=1}^{n} x_i \ln(x_i)$. Assume that $x \in \Delta_n^c$. Then,*

$$
\operatorname*{argmin}_{z \in \Delta_n^c} \langle g, z \rangle + \frac{1}{\alpha} B_{\mu_3}(x, z) = \{z^*\},
$$

*where $z^*$ is given by*

$$
z_i^* = \max\{x_i e^{-\alpha(g_i + \mu^*)}, c\}
$$

*and $\mu^*$ is the unique solution of*

$$
\sum\limits_{j=1}^{n} \max\{x_i e^{-\alpha(g_i + \mu)}, c\} = 1.
$$

*Furthermore, $\mu^*$ can be found using bisection.*

*Proof.* Note that as long as there exists an index $i$ such that

$$
x_i e^{-\alpha(g_i + \mu^*)} > c,
$$

the function

$$
\sum\limits_{j=1}^{n} \max\{x_i e^{-\alpha(g_i + \mu)}, c\}
$$

is strictly decreasing in $\mu$. Furthermore,

$$
\sum\limits_{j=1}^{n} \max\{x_i e^{-\alpha(g_i + \mu)}, c\} \xrightarrow{\mu \to -\infty} \infty,
$$

and

$$\sum_{j=1}^{n} \max\{x_i e^{-\alpha(g_i+\mu)}, c\} \xrightarrow{\mu \to \infty} = n \cdot c < 1.$$

Therefore, the value of $\mu^*$ can be efficiently calculated using bisection.

The Lagrangian $L : \mathbb{R}^n \times \mathbb{R}^n_+ \times \mathbb{R}$ is given by

$$L(z, \lambda, \mu) = \langle g, z \rangle + \frac{1}{\alpha} \left( \sum_{i=1}^{n} z_i \ln(z_i) - \sum_{i=1}^{n} x_i \ln(x_i) - \sum_{i=1}^{n} (\ln x_i + 1)(z_i - x_i) \right) + \mu \left( \sum_{i=1}^{n} z_i - 1 \right) + \sum_{i=1}^{n} \lambda_i (c - z_i).$$

Rearranging,

$$L(z, \lambda, \mu) = \sum_{i=1}^{n} \left( \frac{1}{\alpha} \ln \left( \frac{z_i}{x_i} \right) - \frac{1}{\alpha} + g_i + \mu - \lambda_i \right) z_i - \mu - \sum_{i=1}^{n} \lambda_i c + const.$$

The problem is continuous over a compact set, and therefore, by Weierstrass' theorem, a minimum exists. Since the target function is convex, the KKT conditions are sufficient. The KKT conditions are given by

1. $\frac{1}{\alpha} \ln \left( \frac{z_i}{x_i} \right) + g_i + \mu - \lambda_i = 0$ for all $i \in \{1, \ldots, n\}$.

2. $\lambda_j (c - z_j) = 0$ for all $j \in \{1, \ldots, n\}$.

3. $\sum_{j=1}^{n} z_j = 1$.

4. $z_j \geq c$ for all $j \in \{1, \ldots, n\}$.

5. $\lambda \geq 0$.

The KKT conditions are fulfilled by $(z^*, \lambda^*, \mu^*)$, where $z^*$ and $\mu^*$ are as defined above, and $\lambda^*$ is given by

$$\lambda_j^* = \begin{cases} 0, & z_j^* > c \\ \dfrac{\ln \left( \dfrac{c}{x_i e^{-\alpha(g_i+\mu)}} \right)}{\alpha}, & \text{otherwise} \end{cases}.$$

If $z_j^* > c$, then $\lambda_j^* \geq 0$ trivially. Otherwise, by our choice of $z_j^*$, it follows that

$$x_i e^{-\alpha(g_i+\mu)} \leq c,$$

and hence that $\lambda_j^* \geq 0$. □

**Lemma E.4.** *Let $\mu_1 : \Delta_{n,k}^c \to \mathbb{R}$ be defined as $\mu_1(x) = \frac{1}{2} \|x\|^2$, and $c < \min\{0.5, 1/n\}$. Then,*

$$\operatorname*{argmin}_{z \in \Delta_{n,k}^c} \langle g, z \rangle + \frac{1}{\alpha} B_{\mu_1}(x, z) = \{z^*\},$$

*where $z^*$ is given by*

$$z_j^* = \min\{\max\{x_j - \alpha(g_j + \mu^*), c\}, 1 - c\}.$$

*and $\mu^*$ is the unique solution of*

$$\sum_{j=1}^{n} \min\{\max\{x_j - \alpha(g_j + \mu^*), c\}, 1 - c\} = k.$$

*Furthermore, $\mu^*$ can be found using bisection.*

*Proof.* Note that as long as there exists an index $j$ such that

$$c < x_j - \alpha \left( g_j + \mu^* \right) < 1 - c,$$

the function

$$\sum_{j=1}^{n} \min\{\max\{x_j - \alpha \left( g_j + \mu^* \right), c\}, 1 - c\}$$

is strictly decreasing in $\mu$. Furthermore, the function is continuous, and

$$\sum_{j=1}^{n} \min\{\max\{x_j - \alpha \left( g_j + \mu^* \right), c\}, 1 - c\} \xrightarrow{\mu \to -\infty} n \cdot c < 1 \leq k,$$

and

$$\sum_{j=1}^{n} \min\{\max\{x_j - \alpha \left( g_j + \mu^* \right), c\}, 1 - c\} \xrightarrow{\mu \to \infty} n \cdot (1 - c) > n - 1 \geq k.$$

Therefore, a solution $\mu^*$ exists and can be found using bisection, proving the latter part of the lemma.

To see that the proposed $z^*$ is indeed the optimal solution, note that

$$B_{\mu_1}(z, x) = \frac{1}{2} \|z - x\|^2.$$

Therefore,

$$\langle g, z \rangle + \frac{1}{\alpha} B_{\mu_1}(x, z) = \langle g, z \rangle + \frac{1}{2\alpha} \|z - x\|^2.$$

The Lagrangian $L : \mathbb{R}^n \times \mathbb{R}_+^n \times \mathbb{R}_+^n \times \mathbb{R}$ is given by

$$L(z, \lambda, \eta, \mu) = \langle g, z \rangle + \frac{1}{2\alpha} \|z - x\|^2 + \sum_{i=1}^{n} \lambda_i (c - z_i) + \sum_{i=1}^{n} \eta_i (z_i - 1 + c) + \mu \left( \sum_{i=1}^{n} z_i - k \right).$$

Rearranging,

$$L(z, \lambda, \eta, \mu) = \sum_{i=1}^{n} \left( \frac{1}{2\alpha} z_i^2 + \left( g_i - \frac{x_i}{\alpha} - \lambda_i + \eta_i + \mu \right) z_i \right) + \sum_{i=1}^{n} c \cdot \lambda_i + \sum_{i=1}^{n} (c-1) \cdot \eta_i + k \cdot \mu + \frac{1}{2\alpha} \|x\|^2.$$

The function is continuous and the feasible set is compact, and hence, by Weierstrass' theorem, a minimizer exists. The feasible set and target function are convex, and therefore the KKT conditions are sufficient for optimality.

The KKT conditions are given by

1. $\frac{1}{\alpha} z_j + g_i - \frac{x_i}{\alpha} - \lambda_i + \eta_i + \mu = 0$ for all $j \in \{1, \ldots, n\}$.

2. $\lambda_j(c - z_j) = 0$ for all $j \in \{1, \ldots, n\}$.

3. $\eta_j(z_j - 1 + c) = 0$ for all $j \in \{1, \ldots, n\}$.

4. $\sum_{j=1}^{n} z_j = k$.

5. $z_j \geq c$ for all $j \in \{1, \ldots, n\}$.

6. $z_j \leq 1 - c$ for all $j \in \{1, \ldots, n\}$.

7. $\lambda \geq 0$.

8. $\eta \geq 0$.

The vectors $(z^*, \lambda^*, \eta^*, \mu^*)$ are a solution to the KKT system, where $z^*$ and $\mu^*$ were defined above, $\lambda^*$ is given by

$$\lambda_j^* = \begin{cases} 0, & z_j^* > c \\ \dfrac{c - x_j}{\alpha} + g_j + \mu^*, & \text{otherwise} \end{cases},$$

and $\eta^*$ is given by

$$\eta_j^* = \begin{cases} 0, & z_j^* < 1 - c \\ \dfrac{x_j - 1 + c}{\alpha} - g_j - \mu^*, & \text{otherwise} \end{cases}.$$

$\lambda_j^* \geq 0$ trivially when $z_j^* > c$. When $z_j^* = c$, it implies

$$x_j - \alpha \left(g_j + \mu^*\right) \leq c,$$

and hence

$$\lambda_j^* = \frac{c - x_j}{\alpha} + g_j + \mu^* \geq 0.$$

Likewise, $\eta_j^* \geq 0$ trivially when $z_j^* < 1 - c$. When $z_j^* = 1 - c$, it implies

$$x_j - \alpha \left(g_j + \mu^*\right) \geq 1 - c,$$

and hence

$$\eta_j^* = \frac{x_j - 1 + c}{\alpha} - g_j - \mu^* \geq 0.$$

$\square$

**Lemma E.5.** *Let* $\mu_2 : \Delta_{n,k}^c \to \mathbb{R}$ *be defined as* $\mu_2(x) = -\sum_{i=1}^n \ln(x_i)$. *Assume that* $c < \min\{0.5, 1/n\}$. *Then,*

$$\operatorname*{argmin}_{z \in \Delta_n^c} \langle g, z \rangle + \frac{1}{\alpha} B_{\mu_2}(x, z) = \{z^*\},$$

*where* $z^*$ *is given by*

$$z_i^* = \min\{\max\{\frac{1}{1/x_i + \alpha\left(g_i + \mu^*\right)}, c\}, 1 - c\}$$

*and* $\mu^*$ *is the unique solution of*

$$\sum_{i=1}^n \min\{\max\{\frac{1}{1/x_i + \alpha\left(g_i + \mu\right)}, c\}, 1 - c\} = k$$

*on* $\left(\max_j -\dfrac{1}{\alpha x_j} - g_j, \infty\right)$. *Furthermore,* $\mu^*$ *can be found using bisection.*

*Proof.* Note that as long as there exists an index $j$ such that

$$c < \frac{1}{1/x_i + \alpha\left(g_i + \mu^*\right)} < 1 - c,$$

the function

$$\sum_{j=1}^n \min\{\max\{x_j - \alpha\left(g_j + \mu^*\right), c\}, 1 - c\}$$

is strictly decreasing in $\mu \in \left(\max_j -\dfrac{1}{\alpha x_j} - g_j, \infty\right)$. Furthermore, the function is continuous in $\left(\max_j -\dfrac{1}{\alpha x_j} - g_j, \infty\right)$, and

$$\sum_{j=1}^n \min\{\max\{x_j - \alpha\left(g_j + \mu^*\right), c\}, 1 - c\} \xrightarrow{\mu \to \left(\max_j -\frac{1}{\alpha x_j} - g_j\right)^+} n \cdot (1 - c) < n - 1 \geq k,$$

and

$$\sum_{j=1}^{n} \min\{\max\{x_j - \alpha\left(g_j + \mu^*\right), c\}, 1 - c\} \xrightarrow{\mu \to \infty} n \cdot c < 1 \le k.$$

Therefore, a solution $\mu^*$ exists and can be found using bisection, proving the latter part of the lemma.

Note that

$$B_{\mu_2}(x, z) = -\sum_{i=1}^{n} \ln(z_i) + \sum_{i=1}^{n} \ln(x_i) + \sum_{i=1}^{n} \frac{z_i}{x_i} - n.$$

Therefore, the Lagrangian $L : \mathbb{R}^n \times \mathbb{R}^n_+ \times \mathbb{R}^n_+ \times \mathbb{R}$ is given by

$$L(z, \lambda, \eta, \mu) = \langle g, z \rangle + \frac{1}{\alpha}\left(-\sum_{i=1}^{n} \ln(z_i) + \sum_{i=1}^{n} \frac{z_i}{x_i}\right) + \sum_{i=1}^{n} \lambda_i(c - z_i) + \sum_{i=1}^{n} \eta_i(z_i - 1 + c) + \mu\left(\sum_{i=1}^{n} z_i - 1\right) + const.$$

Rearranging,

$$L(z, \lambda, \eta, \mu) = \sum_{i=1}^{n}\left(\left(g_i + \frac{1}{\alpha x_i} - \lambda_i + \eta_i + \mu\right)z_i - \frac{1}{\alpha}\ln(z_i)\right) + const$$

The feasible set is compact and the target function is continuous, and therefore, by Weierstrass' theorem, a minimizer exists. Since the problem is convex, the KKT conditions are suffcient for optimality. The KKT conditions are given by

1. $g_i + \dfrac{1}{\alpha x_i} - \lambda_i + \eta_i + \mu - \dfrac{1}{\alpha z_i} = 0$ for all $j \in \{1, \ldots, n\}$.

2. $\lambda_j(c - z_j) = 0$ for all $j \in \{1, \ldots, n\}$.

3. $\eta_j(z_j - 1 + c) = 0$ for all $j \in \{1, \ldots, n\}$.

4. $\sum_{j=1}^{n} z_j = k$.

5. $z_j \ge c$ for all $j \in \{1, \ldots, n\}$.

6. $z_j \le 1 - c$ for all $j \in \{1, \ldots, n\}$.

7. $\lambda \ge 0$.

8. $\eta \ge 0$.

The vectors $(z^*, \lambda^*, \eta^*, \mu^*)$ are a solution to the KKT system, where $z^*$ and $\mu^*$ were defined above, $\lambda^*$ is given by

$$\lambda_j^* = \begin{cases} 0, & z_j^* > c \\ g_j + \dfrac{1}{\alpha x_j} + \mu^* - \dfrac{1}{\alpha c}, & \text{otherwise} \end{cases},$$

and $\eta^*$ is given by

$$\eta_j^* = \begin{cases} 0, & z_j^* < 1 - c \\ \dfrac{1}{\alpha(1 - c)} - g_j - \dfrac{1}{\alpha x_j} - \mu^*, & \text{otherwise} \end{cases}.$$

Note that since $c < 0.5$, $z_j^* = c$ and $z_j^* = 1 - c$ are mutually exclusive, and therefore at least one of $\lambda_j^*, \eta_j^*$ is zero.

If $z_j^* > c$, $\lambda_j^* \ge 0$ trivially. Otherwise, by the choice of $z_j^*$,

$$\frac{1}{1/x_i + \alpha(g_i + \mu^*)} \le c,$$

and therefore

$$\lambda_j^* = g_j + \frac{1}{\alpha x_j} + \mu^* - \frac{1}{\alpha c} \ge 0.$$

If $z_j^* < 1 - c$, then $\eta_j^* \geq 0$ trivially. Otherwise, by the choice of $z_j^*$,

$$\frac{1}{1/x_i + \alpha \left(g_i + \mu^*\right)} \geq 1 - c,$$

and therefore

$$\eta_j^* = \frac{1}{\alpha \left(1 - c\right)} - g_j - \frac{1}{\alpha x_j} - \mu^* \geq 0.$$

$\square$

**Lemma E.6.** *Let $\mu_3 : \Delta_{n,k}^c \to \mathbb{R}$ be defined as $\mu_3(x) = \sum\limits_{i=1}^{n} x_i \ln(x_i)$. Assume that $x \in \Delta_{n,k}^c$ and $c < \min\{0.5, 1/n\}$. Then,*

$$\underset{z \in \Delta_n^c}{\operatorname{argmin}} \langle g, z \rangle + \frac{1}{\alpha} B_{\mu_3}(x, z) = \{z^*\},$$

*where $z^*$ is given by*

$$z_i^* = \min\{\max\{x_i e^{-\alpha(g_i + \mu^*)}, c\}, 1 - c\}$$

*and $\mu^*$ is the unique solution of*

$$\sum_{j=1}^{n} \min\{\max\{x_i e^{-\alpha(g_i + \mu^*)}, c\}, 1 - c\} = k.$$

*Furthermore, $\mu^*$ can be found using bisection.*

*Proof.* Note that as long as there exists an index $i$ such that

$$c < x_i e^{-\alpha(g_i + \mu^*)} < 1 - c,$$

the function

$$\sum_{j=1}^{n} \min\{\max\{x_i e^{-\alpha(g_i + \mu^*)}, c\}, 1 - c\}$$

is strictly decreasing in $\mu$. Furthermore,

$$\sum_{j=1}^{n} \max\{x_i e^{-\alpha(g_i + \mu)}, c\} \xrightarrow{\mu \to -\infty} n \cdot (1 - c) > n - 1,$$

and

$$\sum_{j=1}^{n} \max\{x_i e^{-\alpha(g_i + \mu)}, c\} \xrightarrow{\mu \to \infty} = n \cdot c < 1.$$

Therefore, the value of $\mu^*$ can be efficiently calculated using bisection.

Note that

$$B_{\mu_3}(x, z) = \sum_{i=1}^{n} z_i \ln(z_i) - \sum_{i=1}^{n} x_i \ln(x_i) - \sum_{i=1}^{n} \left(\ln(x_i) + 1\right) \left(z_i - x_i\right)$$

$$= \sum_{i=1}^{n} z_i \ln(z_i) - \sum_{i=1}^{n} \left(\ln(x_i) + 1\right) z_i + \sum_{i=1}^{n} x_i$$

$$= \sum_{i=1}^{n} z_i \ln\left(\frac{z_i}{x_i}\right) - \sum_{i=1}^{n} z_i + \sum_{i=1}^{n} x_i.$$

Therefore, the Lagrangian $L : \mathbb{R}^n \times \mathbb{R}_+^n \times \mathbb{R}_+^n \times \mathbb{R}$ is given by

$$L(z, \lambda, \eta, \mu) = \langle g, z \rangle + \frac{1}{\alpha} \left(\sum_{i=1}^{n} z_i \ln\left(\frac{z_i}{x_i}\right) - \sum_{i=1}^{n} z_i + \sum_{i=1}^{n} x_i\right)$$

$$+ \mu \left(\sum_{i=1}^{n} z_i - 1\right) + \sum_{i=1}^{n} \lambda_i \left(c - z_i\right) + \sum_{i=1}^{n} \eta_i \left(z_i - 1 + c\right).$$

Rearranging,

$$L(z, \lambda, \eta, \mu) = \sum_{i=1}^{n} \left( \left( g_i + \mu + \eta_i - \lambda_i - \frac{1}{\alpha} \ln(x_i) - \frac{1}{\alpha} \right) z_i + \frac{1}{\alpha} z_i \ln(z_i) \right) + const.$$

The feasible set is compact and the target function is continuous, and therefore, by Weierstrass' theorem, a minimizer exists. Since the problem is convex, the KKT conditions are suffcient for optimality. The KKT conditions are given by

1. $g_i + \mu + \eta_i - \lambda_i - \frac{1}{\alpha} \ln(x_i) + \frac{1}{\alpha} \ln(z_i) = 0$ for all $j \in \{1, \ldots, n\}$.

2. $\lambda_j(c - z_j) = 0$ for all $j \in \{1, \ldots, n\}$.

3. $\eta_j(z_j - 1 + c) = 0$ for all $j \in \{1, \ldots, n\}$.

4. $\sum_{j=1}^{n} z_j = k$.

5. $z_j \geq c$ for all $j \in \{1, \ldots, n\}$.

6. $z_j \leq 1 - c$ for all $j \in \{1, \ldots, n\}$.

7. $\lambda \geq 0$.

8. $\eta \geq 0$.

The vectors $(z^*, \lambda^*, \eta^*, \mu^*)$ are a solution to the KKT system, where $z^*$ and $\mu^*$ were defined above, $\lambda^*$ is given by

$$\lambda_j^* = \begin{cases} 0, & z_j^* > c \\ \dfrac{\ln\left(\dfrac{c}{x_i e^{-\alpha(g_i + \mu^*)}}\right)}{\alpha}, & \text{otherwise} \end{cases},$$

and $\eta^*$ is given by

$$\eta_j^* = \begin{cases} 0, & z_j^* < 1 - c \\ \dfrac{\ln\left(\dfrac{x_i e^{-\alpha(g_i + \mu^*)}}{1 - c}\right)}{\alpha}, & \text{otherwise} \end{cases}.$$

Note that since $c < 0.5$, $z_j^* = c$ and $z_j^* = 1 - c$ are mutually exclusive, and therefore at least one of $\lambda_j^*, \eta_j^*$ is zero.

If $z_j^* > c$, then $\lambda_j^* \geq 0$ trivially. Otherwise, by the choice of $z_j^*$,

$$x_i e^{-\alpha(g_i + \mu^*)} \leq c.$$

Hence,

$$\lambda_j^* = \frac{\ln\left(\dfrac{c}{x_i e^{-\alpha(g_i + \mu^*)}}\right)}{\alpha} \geq 0.$$

If $z_j^* < 1 - c$, then $\eta_j^* \geq 0$ trivially. Otherwise, by the choice of $z_j^*$,

$$x_i e^{-\alpha(g_i + \mu^*)} \geq 1 - c,$$

and it follows that

$$\eta_j^* = \frac{\ln\left(\dfrac{x_i e^{-\alpha(g_i + \mu^*)}}{1 - c}\right)}{\alpha} \geq 0.$$

$\square$

**Remark E.1.** If $c > 0.5$, the feasible set $\Delta_{n,k}^c$ is always empty. If $c = 0.5$, a feasible solution exists if and only if $n = 2$, in which case the only feasible solution is $z = (0.5, 0.5)$.

# F  SCALABILITY AND PERFORMANCE CONSIDERATIONS

Regarding the practicality and relevance to large-scale applications, below we provide adaptations which may improve the performance of Algorithm 1.

## F.1  THEORY-PRESERVING ADAPTATIONS

1. **Permutation sampling:** Sampling a permutation instead of a subset and adapting the gradient estimator accordingly can reduce the computational cost of gradient evaluation. While we have not yet proven this rigorously, we believe the gradient estimator bounds would still hold. This approach may, however, increase the gradient's variance.

2. **Caching subset evaluations:** Storing values of previously evaluated subsets in a hash table can reduce redundant calculations, especially when the algorithm is close to convergence and repeatedly samples the same subsets.

## F.2  HEURISTIC ADAPTATIONS

1. **Combining sampling techniques:** For large $k$, we can integrate our approach for sampling subsets with expected cardinality $k$ with the method in Pervez et al. (2022). This would yield subsets of exact size $k$, albeit at the cost of slightly biased gradient estimators. Given the design of the sampling process in Pervez et al. (2022), we expect the bias of the gradient estimator to remain low.

2. **Objective evaluation for hyperparameter tuning:** Direct evaluation of the objective function is computationally expensive for large-scale problems. Instead, sampling can be used to estimate the objective efficiently. With upper and lower bounds available, the sampling average converges at a sub-Gaussian rate.

3. **Early stopping criterion:** If a sufficiently good subset is encountered during sampling, we can terminate the process early. This approach avoids unnecessary computation and provides a practical stopping condition for the algorithm.

# G  SYNTHETIC EXPERIMENTS

This section presents synthetic experiments for three Subset Selection tasks: Subset Sum, Unstructured Subset Selection, and Sparse Least Squares. The Subset Sum experiment evaluates the performance of our algorithm on a combinatorial problem without an evident continuous analog. The Unstructured Subset Selection experiments assess our method's effectiveness in a setting devoid of underlying structure. Lastly, the Sparse Least Squares experiment benchmarks our approach against a continuous optimization algorithm suited explicitly for that problem.

## G.1  SUBSET SUM

### G.1.1  SETTING

The first set of synthetic experiments focuses on the *Subset Sum problem*. In this problem, we are given $n$ numbers $Y = \{y_1, \ldots, y_n \in \mathbb{R}\}$, a subset size $k$, and a target value $t \in \mathbb{R}$. The goal is to determine whether there is a subset of elements $k$ from $Y$ such that the sum of the elements equals $t$. The Subset Sum problem is well-known to be NP-Complete. In our synthetic setup, we set $n = 10$, $k = 4$, and $y_i \sim \text{Uniform}[0, 10]$ for all $i \in \{1, \ldots, n\}$. The target $t$ is chosen as the sum of a randomly selected subset of size $k$ from $Y$, ensuring that a solution always exists.

We evaluate the Subset Sum problem under the two settings proposed in this paper: selecting subsets of size $k = 4$ exactly (formulated by ($P_k$)) and selecting subsets of size $k = 4$ in expectation (formulated ($P_B$)). In both cases, for any given subset $C \subseteq Y$, the loss function is defined as:

$$\ell(C) = \left| \left( \sum_{y \in C} y \right) - t \right|.$$

Clearly, $C$ solves the Subset Sum problem if and only if $\ell(C) = 0$.

In the ($P_k$) setting, we use Algorithm 1 with three Bregman divergence variants outlined in Corollary 7.1. For comparison, we implement two baseline stochastic algorithms:

1. **Uniform Sampling:** At each iteration, a subset is chosen uniformly at random, and the best-observed subset is returned.

2. **Stochastic 1-Flip:** It is based on the 1-Flip algorithm, which is discussed, for example, in Szeider (2011). At each step, this method randomly selects two candidate elements, one not in the subset to replace the other in the subset.

For the ($P_B$) model, we only compare with the Uniform Sampling procedure due to the lack of a straightforward analog for the Stochastic 1-Flip algorithm in this setting.

### G.1.2 PARAMETER TUNING

In both settings, experiments have shown that the theoretical values of $\alpha$ derived in Theorem 7.1 and Theorem 7.2 yield a very slow convergence rate when the Bregman divergence base function is $\mu_2(x) = \sum_{i=1}^{n} x_i \ln(x_i)$ or $\mu_3(x) = -\sum_{i=1}^{n} \ln(x_i)$. Consequently, we adopt larger values of $\alpha$:

$$\alpha = n^{2.5} T^{0.5} c^{0.5} \quad \text{for } \mu_2(x), \quad \text{and} \quad \alpha = n^{2.5} T^{0.5} c \quad \text{for } \mu_3(x),$$

where $c$ and $T$ are defined as in Theorem 7.1 and Theorem 7.2. Although these step sizes are not directly supported by our theoretical analysis, they are strongly inspired by it. The Hessians of $\mu_2(x)$ and $\mu_3(x)$ are:

$$\nabla^2 \mu_2(x) = \text{diag}\left(\frac{1}{x_1}, \ldots, \frac{1}{x_n}\right), \quad \nabla^2 \mu_3(x) = \text{diag}\left(\frac{1}{x_1^2}, \ldots, \frac{1}{x_n^2}\right).$$

The bound on $\rho$ derived in Lemma 7.1 and Lemma 7.2 is $O\left((\min_{i=1,\ldots,n} x_i)^{-2}\right)$, relying on the condition:

$$\nabla^2 f^D(x) + O\left(\left(\min_{i=1,\ldots,n} x_i\right)^{-2}\right) I \succeq 0.$$

If we assume the stronger result:

$$\nabla^2 f^D(x) + O(1) \text{diag}\left(\frac{1}{x_1^2}, \ldots, \frac{1}{x_n^2}\right) \succeq 0,$$

then the values of $\rho$ improve to $O\left(\frac{1}{c}\right)$ for $\mu_2(x)$ and $O(1)$ for $\mu_3(x)$. Based on the proofs of Lemma 7.1 and Lemma 7.2, we believe that this heuristic-type assumption is reasonable. Indeed, under this assumption, the chosen $\alpha$ values align with the convergence guarantees in (Zhang and He, 2018, Corollary 3.1).

### G.1.3 RESULTS

The results of the Subset Sum experiments under the ($P_k$) setting are presented in Figure 3 and Table 2. Figure 3 plots the function values and best-observed subsets for two representative instances, while Table 2 summarizes the outcomes of 1000 experiments. Across all configurations, the methods implemented using Algorithm 1 significantly outperform the Uniform Sampling baseline.

Among the three Bregman divergence-based approaches, those utilizing $\mu_1(x) = \|x\|^2$ (Euclid-Squared) and $\mu_2(x) = \sum_{i=1}^{n} x_i \ln x_i$ (Negative Entropy) as the base functions outperform the Stochastic 1-Flip algorithm. However, the approach based on $\mu_3(x) = -\sum_{i=1}^{n} \ln(x_i)$ (Minus Ln) performs slightly worse than Stochastic 1-Flip.

Interestingly, the Negative Entropy method outperforms EuclidSquared in minimizing the function value despite achieving poorer results in identifying the best observed subset. We hypothesize that this discrepancy arises from the higher variability in function values exhibited by the Negative Entropy approach during optimization. This increased fluctuation may lead to sampling subsets from less favorable distributions at specific iterations compared to the EuclidSquared method under equivalent conditions.

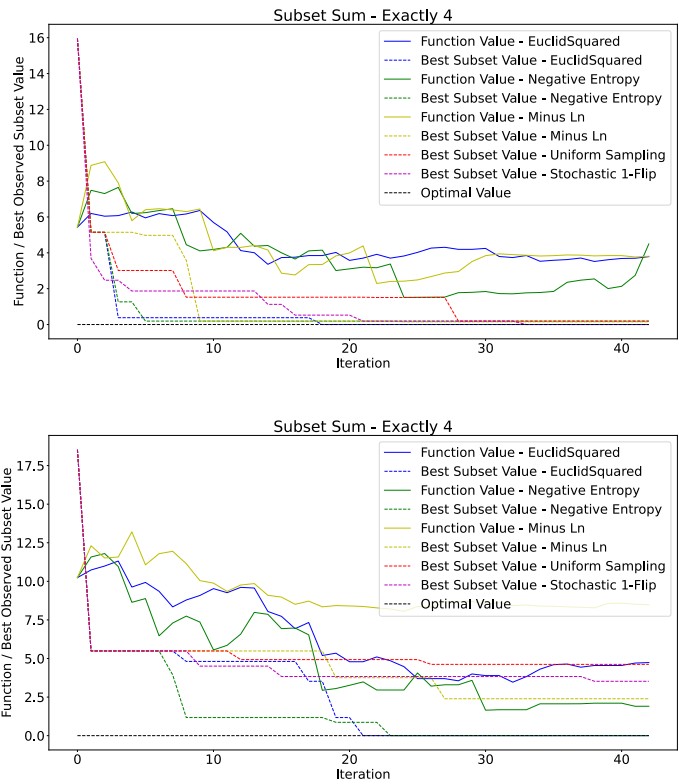

Figure 3: Function values and best observed subset values per iteration – representative samples for ($P_k$) setting.

|  | Best Observed Subset | | Last Iterate Function Value | |
| --- | --- | --- | --- | --- |
|  | Mean | Std | Mean | Std |
| CWR4 - EuclidSquared | **0.145** | 0.270 | 3.607 | 1.161 |
| CWR4 - Negative Entropy | 0.160 | 0.294 | **2.888** | 1.358 |
| CWR - Minus Ln | 0.218 | 0.467 | 4.156 | 2.022 |
| Uniform Sampling | 0.244 | 0.474 | $NA$ | $NA$ |
| Stochastic 1-Flip | 0.208 | 0.313 | $NA$ | $NA$ |

Table 2: Average value of best observed subset and function value for 1000 subset sum instances – ($P_k$) setting. Bold indicates the best in "Mean" column.

The results for the ($P_B$) setting are presented in Figure 4 and Table 3. Figure 4 displays two representative problem instances, while Table 3 provides a summary of the results of 1000 experiments. The three variants of Algorithm 1 outperform the stochastic sampling baseline, with the Minus Ln approach achieving the most significant improvement.

In terms of function value, the Minus Ln approach achieves substantially lower values compared to the Negative Entropy method, with both approaches significantly outperforming the EuclidSquared method. We hypothesize that these differences are primarily attributable to variations in the step size parameter, which may impact the optimization dynamics for each divergence function.

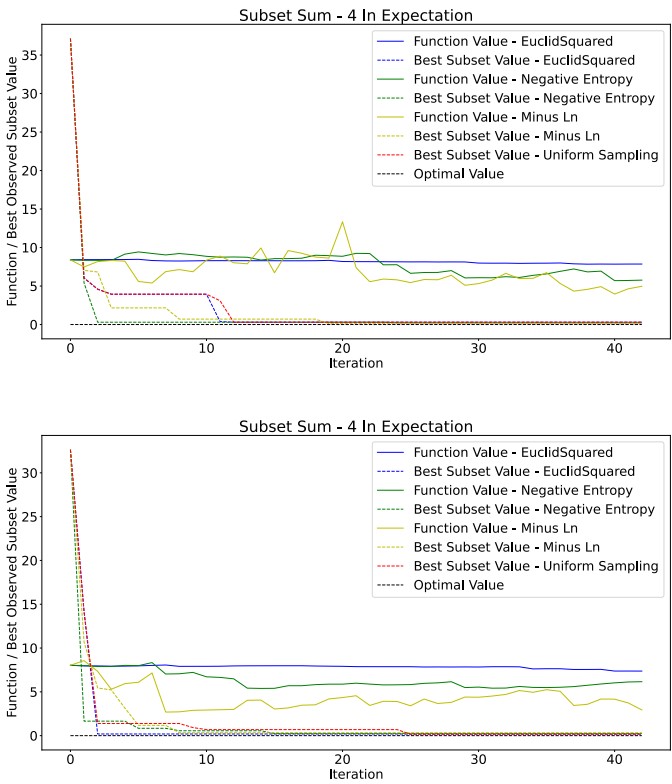

Figure 4: Function Values and Best Observed Subset Values Per Iteration – Representative Samples For ($P_B$) Setting.

|  | Best Observed Subset | | Last Iterate Function Value | |
|---|---|---|---|---|
|  | Mean | Std | Mean | Std |
| HIB4 - EuclidSquared | 0.269 | 0.313 | 7.815 | 1.363 |
| HIB4 - Negative Entropy | 0.267 | 0.307 | 6.432 | 1.287 |
| HIB4 - Minus Ln | **0.239** | 0.265 | **5.760** | 1.895 |
| Uniform Sampling | 0.278 | 0.332 | $NA$ | $NA$ |

Table 3: Average value of best observed subset and function value for 1000 subset sum instances – ($P_B$) setting. Bold indicates the best in "Mean" column.

## G.2 UNSTRUCTURED SUBSET SELECTION

### G.2.1 SETTING

In the unstructured experiment, the value of each subset is assigned independently, with values sampled uniformly at random from the range $[0, 10]$. To ensure that the optimal subset value is 0, we subtract the value of the optimal subset from all subset values. The same algorithms, iteration counts, and step sizes as described in Appendix G.1 are used in this setting.

The unstructured nature of this experiment poses a significant challenge for algorithms that rely on structural patterns, as little to no structure is present.

### G.2.2 RESULTS

The results for the ($P_k$) setting are summarized in Table 4, and two representative instances are visualized in Figure 5. The results show that the Stochastic 1-Flip algorithm, which assumes a strong underlying structure, struggles significantly in this setting. While the variants of Algorithm 1 demonstrate better performance, only the EuclidSquared version marginally outperforms the Uniform Sampling baseline, and even this improvement is negligible.

These findings highlight the inherent difficulty of the unstructured problem, where information about one subset offers no predictive insight into other subsets and very little insight regarding the value of individual elements. This lack of interdependence severely limits the ability of algorithms to exploit any structural advantages, resulting in diminished overall performance.

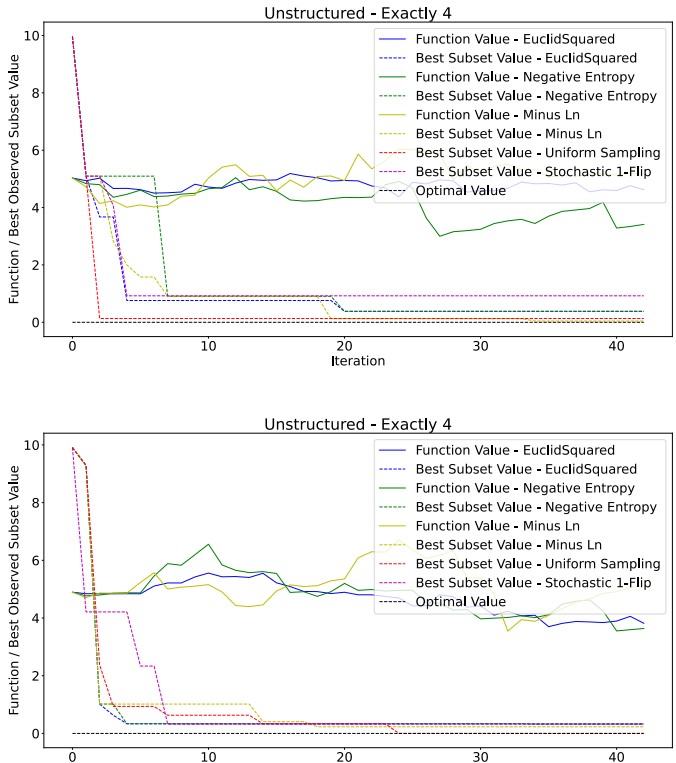

Figure 5: Function values and best observed subset values per iteration – representative samples for ($P_k$) setting.

|  | Best Observed Subset | | Last Iterate Function Value | |
|---|---|---|---|---|
|  | Mean | Std | Mean | Std |
| CWR4 - EuclidSquared | **0.198** | 0.235 | 4.788 | 0.330 |
| CWR4 - Negative Entropy | 0.228 | 0.256 | **4.426** | 0.840 |
| CWR - Minus Ln | 0.247 | 0.278 | 4.781 | 0.524 |
| Uniform Sampling | 0.203 | 0.238 | $NA$ | $NA$ |
| Stochastic 1-Flip | 0.380 | 0.405 | $NA$ | $NA$ |

Table 4: Average value of best observed subset and function value for 1000 subset sum instances – ($P_k$) setting. Bold indicates the best in "Mean" column.

In the ($P_B$) setting, Figure 6 and Table 5 illustrate that all variants of Algorithm 1 struggle to significantly reduce the function value within the allotted number of iterations. The function value remains nearly identical to that of the uniform distribution, and the average best observed subset is similarly close to the one produced by the Uniform Sampling method.

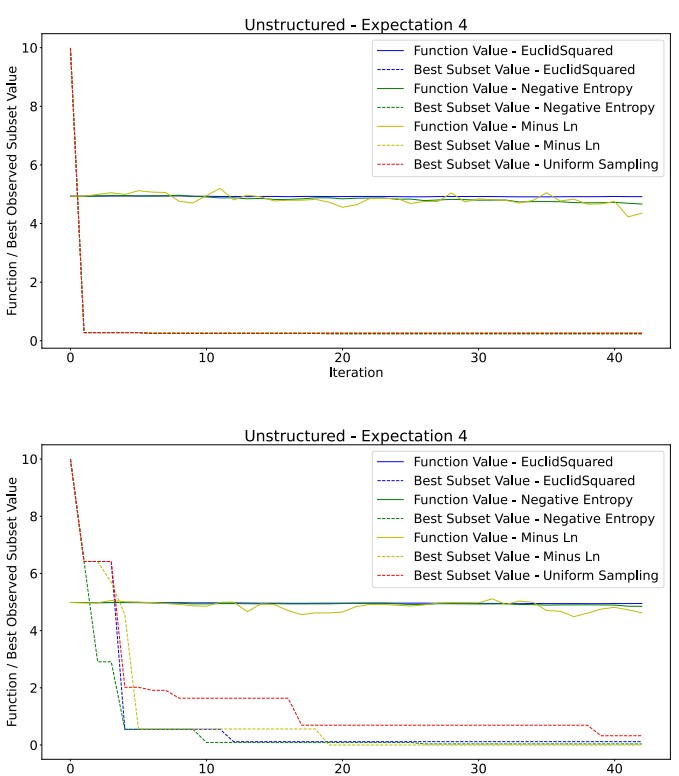

Figure 6: Function values and best observed subset values per iteration – representative samples for ($P_B$) setting.

| | Best Observed Subset | | Last Iterate Function Value | |
|---|---|---|---|---|
| | Mean | Std | Mean | Std |
| HIB4 - EuclidSquared | **0.234** | 0.235 | 4.980 | 0.111 |
| HIB4 - Negative Entropy | 0.240 | 0.244 | 4.952 | 0.155 |
| HIB - Minus Ln | 0.239 | 0.249 | **4.901** | 0.336 |
| Uniform Sampling | 0.238 | 0.246 | $NA$ | $NA$ |

Table 5: Average value of best observed subset and function value for 1000 subset sum instances – ($P_B$) setting. Bold indicates the best in "Mean" column.

### G.3 SPARSE LEAST SQUARES

#### G.3.1 SETTING

The final set of experiments is conducted in a Sparsity-Constrained Least Squares setting, defined as

$$\min_{x \in \mathbb{R}^n} \left\{ \|Ax - b\|^2 \text{ subject to } \|x\|_0 \leq k \right\},$$

where the $\|\cdot\|_0$ "norm", which counts the number of nonzero entries, is given by

$$\|z\|_0 = \sum_{i=1}^{n} 1_{z_i \neq 0}.$$

In each problem instance, we sample the matrix $A \in \mathbb{R}^{8 \times 10}$, where each entry is independently sampled from a standard normal distribution: $A_{i,j} \sim \mathcal{N}(0,1)$. The ground truth vector $\hat{x}$ is generated by first sampling entries $x_i \sim \mathcal{N}(0,1)$ independently, and then setting all but 4 randomly selected entries of $\hat{x}$ to zero. Finally, we set $b = A\hat{x} + \epsilon$, where the noise vector $\epsilon$ has entries sampled independently from $\epsilon_i \sim \mathcal{N}(0, 0.1)$.

We apply all the algorithms and settings discussed in Appendix G.1. Additionally, in the ($P_k$) setting, we also test a projected gradient descent algorithm, with a step size of $\frac{1}{L}$, where $L$ is the Lipschitz constant of $g(x) = \|Ax - b\|^2$. The convergence properties of projected gradient descent in this setting are discussed in Beck and Eldar (2013).

### G.3.2   RESULTS

The results for the ($P_k$) setting are provided in Figure 7 and Table 6. Two of the three variants of Algorithm 1 outperform all other methods, despite the relatively structured nature of the problem, which we believe to be advantageous for an algorithm such as Stochastic 1-Flip.

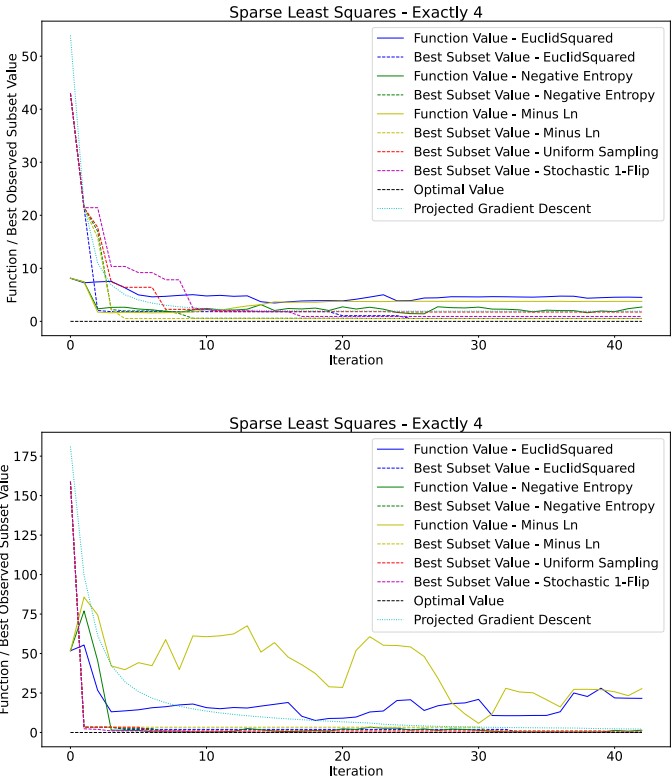

Figure 7: Function values and best observed subset values per iteration – representative samples for ($P_k$) setting.

|                            | Best Observed Subset | | Last Iterate Function Value | |
|----------------------------|--------|--------|--------|--------|
|                            | Mean   | Std    | Mean   | Std    |
| CWR4 - EuclidSquared       | 1.056  | 1.380  | 8.208  | 8.986  |
| CWR4 - Negative Entropy    | **1.026** | 1.505 | **7.493** | 11.311 |
| CWR - Minus Ln             | 1.222  | 1.601  | 10.772 | 10.312 |
| Uniform Sampling           | 1.331  | 1.658  | $NA$   | $NA$   |
| Stochastic 1-Flip          | 1.067  | 1.532  | $NA$   | $NA$   |
| Projected Gradient Descent | 3.690  | 4.438  | $NA$   | $NA$   |

Table 6: Average value of best observed subset and function value for 1000 subset sum instances – ($P_k$) setting. Bold indicates the best in "Mean" column.

In the ($P_B$) setting, the variants of Algorithm 1 outperform the Uniform Sampling approach. Similarly to the results for the ($P_B$) in Appendix G.1, we can see that the best average function value result does not necessarily translate to the best observed subset result. As before, we speculate that this is due to fluctuations in the function value in the Minus Ln approach which are very apparent in the sample instances illustrated in Figure 8.

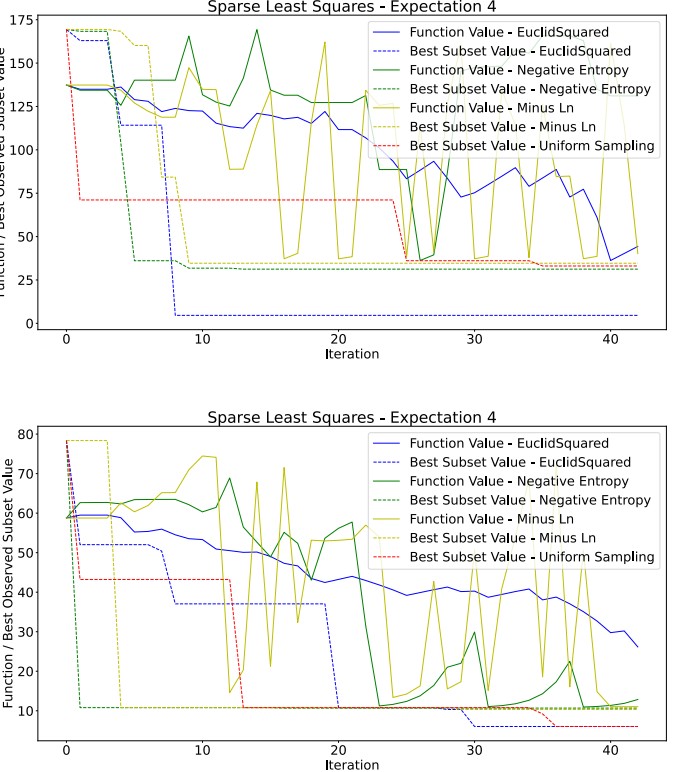

Figure 8: Function values and best observed subset values per iteration – representative samples for ($P_B$) setting.

|                            | Best Observed Subset | | Last Iterate Function Value | |
| -------------------------- | ------ | ------ | ------ | ------ |
|                            | Mean | Std | Mean | Std |
| HIB4 - EuclidSquared       | 8.982 | 9.140 | 34.605 | 27.865 |
| HIB4 - Negative Entropy    | **8.688** | 8.372 | 32.084 | 32.375 |
| HIB4 - Minus Ln            | 10.483 | 10.748 | **28.663** | 30.810 |
| Uniform Sampling           | 10.863 | 9.798 | $NA$ | $NA$ |

Table 7: Average value of best observed subset and function value for 1000 subset sum instances – ($P_B$) setting. Bold indicates the best in "Mean" column.

