# OpenReview forum: "A Stochastic Approach to the Subset Selection Problem via Mirror Descent"
_ICLR.cc/2025/Conference — ICLR 2025 Poster_

### Official Review · Reviewer_dyZD · 2024-11-02

**Soundness:** 3
**Presentation:** 3
**Contribution:** 3
**Rating:** 6
**Confidence:** 4

**Summary:**

This paper presents a novel stochastic optimization algorithm to the *subset selection" problem. The objective of this problem is to minimize a loss on the subsets of an ambient set, and this combinatorial problem is NP-hard.  This paper proposed two stochastic approximations where the optimization variable instead becomes a probabilistic distribution over the subsets.

Compare to previous works in stochastic subset selection problem, this paper only makes little assumption over the loss function, where it is only defined over the discrete subsets and thus not differentiable over some continuous space.

To approach this setting, the authors defined a stochastic gradient, where the derivative is taken over the parameters determining the probability distributions over the subsets. They further show that this stochastic gradient is 1) unbiased, 2) bounded, and 3) relatively convex (an extended notion of strong convexity). This properties allows the author to apply a stochastic mirror descent algorithm and utilize existing analysis in the literature [1] to prove convergence of their algorithm.

Finally, the authors provided a practical application of their method for the transfer learning problem and provided experimental results.

[1] Zhang S, He N. On the convergence rate of stochastic mirror descent for nonsmooth nonconvex optimization. arXiv preprint arXiv:1806.04781. 2018.

**Strengths:**

Overall, this paper is well-written and offers a neat solution to a difficult problem. While there are not any mind-blowing mathematical techniques and the prior literature [1] does quite a bit of heavy lifting, the proposed method has sufficient contribution to warrant a publication.

The biggest strength of this paper is the writing. The authors does a good job at guiding the reader through the various steps of their method and their claims are backed-up by technical results. There are no "fluff" in the paper in the sense that all results contribute meaningfully to the overall picture.

I checked most of the proofs and they all seem to be correct up to constants or typos. I did not check Lemma 7.1 or 7.2, but I presume some variation of those claims must be true. The other proofs all look in good shape (in a first pass), and the techniques generally follow established literature so I don't expect there is any hidden "surprises".

My overall assessment is that paper is not ground-breaking but does the job. It solves a relevant problem with a well-constructed method. While the lack of mathematical novelty stops it from being a great paper, it should solidly pass the threshold for acceptance.

**Weaknesses:**

While the overall quality of the paper is good, there are a few places where the author's intentions could be better clarified,

- The formulation of the mirror descent step in line 4 of Algorithm 1 is non-standard and I suggest the author to add a few lines of math to show how one can arrive at this line from a more standard form of mirror descent.

- Statements of Theorem 5.1 and 5.2 are rather difficult to parse. Initially, it wasn't immediately clear to me if smaller optimality gap $\tau$ implies smaller $c^*$. The author should some discussion to guide the readers on how to better interpret the equation on on 291, and a plot could be nice here. Also, in the proof, line 802 suffers a similar issue and more detailed steps should be provided.

- Section 6 claimed the stochastic gradient is inspired by the REINFORCE RL algorithm. But I cannot see this connection at all. I think it would be better if the authors just directly offer the intuition through a few lines of math instead of wrapping the logic around a gray box.

- The experimental setup with transfer learning may be unfamiliar for many readers and a more gentle introduction to this problem is necessary. I personally had to Google this because the paragraph at the bottom of page 8 is on the shorter side. Also, I am still confused by the role of HPO. The authors should more explicitly state the objectives of these experiments and describe the terminologies for a broader audience.

- The numerical results in Table 1 does not show a significant advantage for the author's approach. But I don't view this as essential to the overall quality of the paper.

Minor errors/typos:
- in Theorem 7.3, should be $(u-l)^2$.
- line 281, should be relaxation in $(P_k^c)$.
- line 673: should be Lemma B.1.1
- line 837, repeated $\in$ symbol

**Questions:**

- Regarding the stochastic relaxation from $(P)$ to $(P_k)$ and $(P_B)$, do you know the optimality gap of this relaxation? If this has already been studied in the literature, it would nice to have a brief discussion over those results.
- Just to make sure, would Lemma B.1 imply $L^* = L^*_k = L^*_B$? Also, I don't think this lemma is necessary for the proof of Theorem 5.1/5.2?
- What is Appendix E for?

---

> ### Author Response · Authors · 2024-11-20
>
> Thank you for your positive and constructive review. We appreciate your thoughtful comments and have worked to address them in the following rebuttal and accompanying revisions to the manuscript.
>
> ### **Nonstandard Mirror Descent Step**
> Thanks for pointing this out, we  reformulated  the Mirror Descent step to a more standard (equivalent) expression, which we hope improves readability.
>
> ### **Statements of Theorems 5.1 and 5.2**
> We updated the statements of these theorems to better reflect the relationship between $\tau$ and $c$, clarifying that a decrease in $\tau$ leads to a decrease in $c$. Additionally, we included a remark to further elucidate this relationship.
>
> ### **Referencing the REINFORCE Estimator**
> Following your suggestion, we have removed the references to the REINFORCE algorithm and replaced them with an alternative intuition, which we believe aligns better with the context of our work – thanks.
>
> ### **Transfer Learning (TL) and Hyperparameter Optimization (HPO)**
> To address your feedback, we have added a brief explanation of TL to the opening paragraph of Section 8.
> Regarding HPO, it is a standard practice used to optimize parameters such as the learning rate during model training. In our case, we optimized $\alpha$ and $\tau$. To ensure fairness, we allowed HPO for the benchmark algorithms as well.
>
> ### **Numerical Results**
> We have conducted additional synthetic experiments, which we hope better demonstrate the advantages of our approach and provide further empirical support for our claims.
>
> ### **Optimality Gap from $(P)$ to $(P_k)$ and $(P_B)$, and from $L^{*}$ to $L_k$ and $L_B$**
> We clarify the relationship between these formulations:
> - The minimum of $(P)$, $L^{*}$, equals the infimum of $(P_k)$, $L_k$, as shown in Lemma B.1.
> - The minimum of $(P_B)$, $L_B$, provides a lower bound on both $L^{*}$ and $L_k$.
>
> However, without additional restrictions on the choice of $\ell$, the optimality gap between $(P_B)$ and $(P)$ can be unbounded. Consider the following adversarial extension of $\ell$ to subsets of cardinality different than $k$ – set $\ell(C) = m$ for a subset $C$ with cardinality different from $k$, and let $m \to -\infty$. In this scenario, any set of weights that assigns positive probability to $C$ results in the problem value becoming a constant fraction of $m$, leading to an unbounded gap. Specifically, this applies when all element weights are restricted to $(0,1)$, as every subset $C$ retains a positive probability of selection.
>
> ### **Inclusion of Lemma B.1**
> While Lemma B.1 is not strictly necessary for the proofs of Theorems 5.1 and 5.2, we believe it serves an important role by providing insight into the transition from $(P)$ to $(P_k)$ and $(P_B)$. For this reason, we propose retaining it in the appendix to preserve its explanatory value. We have removed the sentence between Lemma B.1 and the proof of Theorem 5.1, which incorrectly stated that Lemma B.1 is used in the proof of Theorems 5.1 and 5.2.
>
> ### **Appendix E**
> Appendix E establishes a theoretical foundation for efficiently calculating the Mirror Descent step, which we believe could be valuable for those implementing our algorithm.
>
> Thank you again for your thoughtful feedback and positive evaluation of our work. We are confident that the revisions and clarifications we have made will address your concerns and further strengthen the manuscript.

---

> > ### Comment · Reviewer_dyZD · 2024-11-25
> >
> > Thank you for the detailed response. I find the clarification on TL very helpful and the new numerical results on the synthetic tasks a valuable addition to this paper.
> >
> > I will maintain my score of **6**. And if I could give a more precise score, the revision elevates the paper to a **6+**.

---

> > > ### Author Response · Authors · 2024-11-26
> > >
> > > Thank you.

---

### Official Review · Reviewer_eDJC · 2024-11-03

**Soundness:** 3
**Presentation:** 3
**Contribution:** 2
**Rating:** 6
**Confidence:** 4

**Summary:**

The authors address the subset selection problem, a fundamental task in machine learning and computer science, by introducing a stochastic formulation for selecting minimum-cost subsets in a black-box setting. Here, only the metric value for a subset is accessible, limiting the information available for the selection process. The authors propose a two-stage framework where subset selection is decoupled into an outer selection component and an inner cost evaluation component. The paper parameterizes the subset distribution using a decision variable, optimized via Stochastic Mirror Descent (SMD). The proposed distributions enable constructive, closed-form, unbiased stochastic gradient formulas with favorable convergence guarantees. The approach is empirically validated on a subset selection task involving layer selection in transfer learning, showcasing both theoretical and practical benefits.

**Strengths:**

1. This paper presents an approach to the subset selection problem by proposing a stochastic formulation optimized using Stochastic Mirror Descent (SMD). Traditional subset selection techniques often involve deterministic methods, such as greedy algorithms or combinatorial optimization, which lack the flexibility and exploration capabilities needed for high-dimensional or complex subset spaces.

2. They derive closed-form, unbiased stochastic gradients for the subset distribution parameters, which makes the method practical and computationally efficient, especially compared to approaches requiring approximate gradients.

3. Overall, the paper is well-structured and clear, with a logical progression from the problem formulation to the proposed approach, theoretical analysis, and empirical evaluation.

4. The significance of this work is substantial, both theoretically and practically. The subset selection problem is fundamental in machine learning, impacting areas like feature selection, model compression, neural architecture search, and sensor placement. By proposing a stochastic approach, this paper introduces a new direction for subset selection that emphasizes flexibility, exploration, and computational efficiency, which is especially relevant in high-dimensional or complex problem spaces.

**Weaknesses:**

1. While the layer selection experiment in transfer learning provides an interesting and relevant use case, the empirical evaluation is limited to a single task. Subset selection has a wide range of applications across machine learning, including feature selection, model pruning, sensor placement, and combinatorial optimization. Evaluating the proposed method on a broader range of tasks (as stated above) would strengthen the paper and better demonstrate the method’s generalizability.

2. There are various existing subset selection methods, both deterministic (e.g., greedy algorithms, combinatorial optimization) and stochastic (e.g., Monte Carlo methods, stochastic optimization techniques). The paper lacks a comparison with these methods, which would provide readers with more context on the advantages and disadvantages of the proposed approach and help highlight its unique contributions.

3. The paper does not clearly address how the assumptions of smoothness and favorable structure in the subset metric or cost function impact the theoretical results’ robustness in settings with irregular cost functions. Extending the analysis to cover more realistic conditions or discussing the limitations of these assumptions would provide a clearer understanding of the method’s applicability in diverse practical scenarios.

4. The choice of specific distributions for sampling subsets is not well-justified in the paper. While the authors mention that the distribution parameters are optimized, they do not explain why these particular distributions were chosen or how they might perform compared to other candidate distributions. This choice is critical, as different distributions could have substantial effects on the subset sampling’s efficiency and coverage. A clearer explanation of the rationale behind the selected distributions or a comparison with alternative options would strengthen the argument and clarify the trade-offs involved in the design.

5. Despite the use of stochastic methods, the proposed approach may still involve significant computational costs, particularly when repeatedly sampling and evaluating subsets over many iterations. A more thorough discussion on the scalability, computational complexity, and potential approaches to mitigate these challenges would make the paper more practically relevant for large-scale applications.

**Questions:**

1. What are the specific properties of the chosen distributions that make them suitable for subset selection? Are these distributions known to have properties that benefit the subset selection problem?

2. How does the approach handle non-smooth or noisy subset metrics? Would there be significant degradation in performance, or could the method be adapted to handle irregular cost functions?

3. Can you discuss the relative benefits of each stochastic formulation (fixed vs. expected cardinality)? Could the authors discuss specific scenarios where one formulation might be preferable over the other?

4. What potential approximations could make this approach more feasible for large-scale problems? Have the authors considered any approximation techniques, such as subsampling or early stopping, that could reduce the computational cost? Discussing potential modifications or approximations would provide insights into how the method could be adapted for resource-constrained environments.

---

> ### Author Response · Authors · 2024-11-20
>
> **Due to the maximum comment length limit, we split this comment into two parts.**
>
> Thank you for your positive and detailed review. We appreciate your thoughtful comments and hope that this rebuttal, alongside the accompanying revisions, will address your concerns and further clarify our contributions.
>
> ### **Experiments**
> We added three sets of synthetic experiments comparing our approach to Monte Carlo, 1-Flip and Projected Gradient Descent methods (the latter is included in one of the three setups). These experiments complement the Transfer Learning (TL) experiment: while the TL experiment demonstrates the applicability of Subset Selection to diverse learning tasks, the new synthetic experiments highlight the performance and robustness of our Subset Selection algorithm in various scenarios.
>
> ### **The Effect of Favorable Structure**
> As shown in the new synthetic experiments, favorable structure strongly influences the convergence rate of our algorithm. This effect can also be seen in the gradient estimator formula: if a sampled subset $C$ incurs a high loss, the probabilities of the elements in $C$ being selected decreases, and vice versa. This behavior reflects an implicit assumption of our algorithm: that the subset loss provides meaningful information about the influence of its elements.
>
> At a minimum, for any element $e \in C$, the subset loss $\ell(C)$ provides information about one subset containing $e$ (specifically $C$ itself). When this assumption holds more strongly—e.g., $\ell(C)$ provides insights into losses of other subsets sharing elements with $C$—the algorithm’s performance improves. We have included a concise version of this discussion in the paper.
>
> ### **Subset Distribution Choices**
> The distribution classes we use meet three key criteria:
> - They can represent the distribution that selects the optimal subset with probability 1 (for the weighted choice without replacement distribution, this holds in limit due to the open feasible set).
> - They allow efficient computation of gradient estimators.
> - We establish favorable properties for the induced optimization problems, such as bounds on the singular values of the Hessian and second-moment bounds on the gradient estimator.
>
> It seems plausible that similar results could be achieved for any distribution satisfying these criteria, although further investigation would be required to verify it.
>
> ### **Scalability and Performance Considerations**
> Regarding the practicality and relevance to large-scale applications, below we provide detailed performance enhancing adaptations.
>
> #### **Theory-Preserving Adaptations**:
> - Choice without replacement:
>    Sampling a permutation instead of a subset and adapting the gradient estimator accordingly can reduce the computational cost of gradient evaluation. While we have not yet proven this rigorously, we believe the gradient estimator bounds would still hold. This approach may, however, increase the gradient’s variance.
>
> - Caching subset evaluations:
>    Storing values of previously evaluated subsets in a hash table can reduce redundant calculations, especially when the algorithm is close to convergence and repeatedly samples the same subsets.
>
> #### **Heuristic Adaptations**:
> - Combining sampling techniques:
>    For large $k$, we can integrate our approach for sampling subsets with expected cardinality $k$ with the method in [1]. This would yield subsets of exact size $k$, albeit at the cost of slightly biased gradient estimators. Given the design of the sampling process in [1], we expect the bias of the gradient estimator to remain low.
>
> - Objective evaluation for hyperparameter tuning:
>    Direct evaluation of the objective function is computationally expensive for large-scale problems. Instead, sampling can be used to estimate the objective efficiently. With upper and lower bounds available, the sampling average converges at a sub-Gaussian rate.
>
> - Early stopping criterion:
>    If a sufficiently good subset is encountered during sampling, we can terminate the process early. This approach avoids unnecessary computation and provides a practical stopping condition for the algorithm.
>
> We aim to incorporate some version of these considerations into the paper, balancing the available space with their importance.

---

> ### Author Response · Authors · 2024-11-20
>
> ### **Noisy Subset Metrics**
> The impact of noisy subset metrics is an excellent question that warrants further investigation. Based on our understanding, the convergence results should hold under the following conditions:
> 1. The gradient estimators remain unbiased despite the noise. We expect this to hold when the noise is independent of the sampling process, though this requires further investigation.
> 2. The noise is bounded, ensuring the sampled subset values remain within known lower and upper bounds.
>
> ### **Fixed vs. Expected Cardinality**
> Broadly speaking, the choice between fixed and expected cardinality depends on whether cardinality is a strict constraint or a recommendation. For example:
> - In a Subset Sum problem, subsets of size $k-2$ are not feasible, making fixed cardinality essential.
> - In feature selection, subsets with approximately $k$ elements may suffice, making expected cardinality more practical.
>
> Moreover, the expected cardinality setting can encode fixed cardinality by assigning a loss slightly above the upper bound to subsets with undesired sizes. However, this can slow convergence due to the prevalence of low-quality samples.
>
> Lastly, the expected cardinality setting scales much better with $k$, offering significant advantages in large-scale applications. We have included a concise discussion of this comparison in the paper.
>
> Thank you again for your constructive feedback and for highlighting these important considerations. We hope our response and the revisions to the paper address your concerns.
>
> ### **References**
> [1] Adeel Pervez, Phillip Lippe, and Efstratios Gavves. Scalable subset sampling with neural conditional Poisson networks. *In The Eleventh International Conference on Learning Representations,* 2022.

---

### Official Review · Reviewer_cTYy · 2024-11-04

**Soundness:** 3
**Presentation:** 3
**Contribution:** 3
**Rating:** 8
**Confidence:** 3

**Summary:**

This paper studies the subset selection problem. A two stochastic formulations of the
problem are proposed, as well as a computable approximations, and solved via mirror descent, and
a $1/\sqrt{T}$ rate of convergence to a stationary point is proven. The methods are tested
empirically in a transfer learning task.

**Strengths:**

- It is rather surprising to me that the approach is seemingly able to
get around making assumptions on the underlying structure of the losses that
that one would see in the combinatorial bandits literature --- ie, that
the losses take the form $\langle\ell_t, w\rangle= \sum_i \ell_{ti}\mathbb{I}(w=1)$, whereas in this
work it is possible to have losses defined in such a way that every possible subset maps to independent values.

- I really like the idea of second approach, in which the condition that $\\|w\\|_1\le k$ is relaxed
to $\\|w\\|_1\le k$ *on average*, in exchange for a more readily computable update

- The paper is generally well-written and well explained

**Weaknesses:**

- Lines 33/34: we make *no assumptions* regarding the loss function other than having lower and upper bounds
    - This is not true; the results rely on relative weak convexity of $f$ wrt $\mu$, and moreover any assumptions made about $\mu$
    will imply certain restrictions on the curvature of $f$ as well. For instance, the stationarity measure is related to
    the more traditional one for $\mu$ with Lipschitz gradients, and hence the results are not necessarily
    meaningful unless $f$ is RWC wrt some smooth $\mu$, which implicitly limits what $f$ can be
    - The approach critically relies on the approximations to the stochastic problems, and choosing a valid approximation constant
    seems to require prior knowledge of $\max_C \ell(C)$ and $\min_C \ell(C)$. So we must not only assume that $\ell$ is bounded, but
    that the bounds are *known* to the user. This is arguably a very strong assumption to make.

- The experiments feature an interesting *application* of the subset selection problem, but given that this paper is about a new approach to a rather fundamental problem, it would have been more illuminating to see results in more standard subset selection problems such as Vertex Cover and comparisons against existing methods. As it stands, the experiments currently in the paper do not shed any light on how well the new approach works for subset selection, because it's not clear to me how important subset selection really is for transfer learning.

- Theorem 7.3 seems unnecessary --- it appears to just be a straight-forward application of Azuma-Hoeffding.

- There is a long history of using a smoothing of the loss function to relax convexity assumptions; perhaps there should be more discussion on this front in the related work

### Minor Details
- Line 199: This is a very atypical way to write the mirror descent update; is there
  a reason the update is not written in terms of the Bregman divergence?

- It looks like you might be using \Cref in some of your appendix names, you might want to consider using something more based on the standard \ref call, like "Proofs of Section~ \ ref{sec:gradientestimators}", in these places. This will still put an in-text reference with hyperlink, but now the index in the pdf table of contents will correctly say "Proofs of Section 7".

**Questions:**

- As mentioned above, the approach seems to work even when the loss is defined in such a way that
every subset maps to a distinct and unrelated loss. This suggests that the problem setting should be significantly harder
than e.g. the combinatorial bandits setting, where we just need to find all the "best" indices. In fact, this difficulty seems to be reflected directly in the upperbound: the numerator is a factor of $n^2$ larger than what is attainable in the combinatorial bandits setting ($\sqrt{n}$). Do you expect that this is the optimal dependence?

---

> ### Author Response · Authors · 2024-11-20
>
> Thank you for your positive and thoughtful review. We appreciate your feedback and aim to address your questions and concerns both in this rebuttal and in the accompanying revisions to the paper.
>
> 1. **Assumptions on $\ell (\cdot)$:**
>    Please first note that the requirement that $f  (\cdot)$ will be RWC with respect to $\mu$ imposes no restrictions on $\ell (\cdot)$ itself, as $f (\cdot)$ remains RWC for any choice of $\ell$ and any $1$-strongly convex $\mu$ (as noted in Remark 3.1). We understand that this can be described better and so we revise accordingly that the RWC constant $\rho$ does depend on $\max\limits_{C \in \mathcal{C}} \ell(C)$.
>    Regarding the bounds on $\ell(C)$, our approach indeed requires knowledge of some lower and upper (not necessarily tight) bounds on $\ell(C)$, which might be nontrivial depending on setup – we have revised accordingly, thanks.
>
> 2. **Experiments:**
>    We appreciate and agree with your suggestion regarding the experiments. New synthetic experiments designed to clearly demonstrate our approach in more classical problems were added. Due to space constraints, these experiments are included in the appendix, and we refer the readers to the relevant section in the appendix from the Experiments section.
>
> 3. **Theorem 7.3:**
>    We fully agree that the proof of Theorem 7.3 is a direct application of Hoeffding’s inequality – Its inclusion is not because of the theoretical derivation, rather, it serves an important role in the overall narrative of the paper. Specifically, our intended contribution can be summarized as follows:
>    - Subset Selection can be transformed into an equivalent stochastic optimization problem.
>    - The stochastic problem can be approximated arbitrarily well.
>    - We propose an algorithm for the approximate stochastic problem that converges to solutions satisfying necessary optimality criteria at a known rate.
>    - Any solution to the approximate stochastic problem can be used to efficiently sample subsets with values arbitrarily close to---or better than---the stochastic solution.
>
>    In this context, Theorem 7.3 implements the fourth step, bridging the stochastic problem and the original subset selection task. We can further emphasize its purpose in the paper if you believe that it is necessary, but we see it as an integral part of the main text and therefore, think it should not be omitted.
>
> 4. **Smoothing Techniques:**
>    Thanks for pointing this out. We added a brief mention of alternative smoothing techniques, but unfortunately, due to space limitations, we have no space to elaborate further.
>
> 5. **Lower Bound Dependence on $n$:**
>    Regarding your question on the lower bound dependence on $n^{2.5}$---this is an excellent question. Unfortunately, we do not have a definitive answer at this time. The question becomes even more complex because our algorithm converges to necessary optimality conditions (which is the standard type of result in nonconvex optimization) rather than to global optimality.
>
> 6. **Minor Comments:**
>    We agree with both of your minor comments and have addressed the issues accordingly.
>
> Thank you again for your feedback and constructive comments. We hope these changes address your concerns and further clarify the contributions of our work.

---

> > ### Comment · Reviewer_cTYy · 2024-11-26
> >
> > Thanks for the detailed response!
> >
> > I might be missing something obvious, but I don't see how remark 3.1 fixes my concern.
> > For instance, suppose $\mu(x)=x^2$, then $f(x)=-ax^2$ is $(\rho,\mu)$  RWC only for $a<\rho$. So the assumption that $f$ is $(\rho,\mu)$ RWC implicitly puts restrictions on what $f$ could possibly be, it is not accurate to say that this paper places absolutely no assumptions on the losses

---

> > > ### Author Response · Authors · 2024-11-26
> > >
> > > Thank you for your question. I believe that the missing piece here is that $f(\cdot)$ is not an arbitrary function we receive as input – it is constructed as an expectation over the values of $\ell(\cdot)$, where the expectation is with respect to the distribution induced by the decision variables.
> > >
> > > It is the distribution family of choice that makes $f(\cdot)$ relatively weakly convex with a certain constant, and the bounds of $\ell(\cdot)$ affect this constant. We show it on a small example below.
> > >
> > > This example considers the choice without replacement scenario, with $n=3$, $k=2$, and
> > > $\ell(\cdot)$ defined as $\ell(\\{1,2\\}) = 0$, $\ell(\\{1,3\\}) = 1$, $\ell(\\{2,3\\}) = 2$.
> > >
> > > So, the probability of choosing the subset $\\{0,1\\}$ given a weights vector $(x_1,x_2,x_3)^T$ is:
> > >
> > > $x_1 \cdot \frac{x_2}{1-x_1} + x_2 \cdot \frac{x_1}{1-x_2}$.
> > >
> > > Similarly, the probabilities of choosing the subsets $\\{0,2\\}$ and $\\{1,2\\}$ are:
> > >
> > > $x_1 \cdot \frac{x_3}{1-x_1} + x_3 \cdot \frac{x_1}{1-x_3}$,
> > >
> > > and
> > >
> > > $x_2 \cdot \frac{x_3}{1-x_2} + x_3 \cdot \frac{x_2}{1-x_3}$,
> > >
> > > respectively.
> > >
> > > Therefore, the expectation function $f(\cdot)$ is given by:
> > >
> > > $f(x_1,x_2,x_3) = \left(x_1 \cdot \frac{x_2}{1-x_1} + x_2 \cdot \frac{x_1}{1-x_2}\right) \cdot 0 + \left(x_1 \cdot \frac{x_3}{1-x_1} + x_3 \cdot \frac{x_1}{1-x_3}\right) \cdot 1 + \left(x_2 \cdot \frac{x_3}{1-x_2} + x_3 \cdot \frac{x_2}{1-x_3}\right)\cdot 2$.
> > >
> > > Some calculations show that the Hessian is given by:
> > >
> > > $\begin{bmatrix}
> > > \frac{2x_{2}}{\left(1 - x_{0}\right)^{3}} & 0 & \frac{x_{2}}{\left(1 - x_{2}\right)^{2}} + \frac{1}{1 - x_{2}} + \frac{x_{0}}{\left(1 - x_{0}\right)^{2}} + \frac{1}{1 - x_{0}} \\\\
> > > 0 & \frac{4x_{2}}{\left(1 - x_{1}\right)^{3}} & 2 \left(\frac{1}{\left(1 - x_{1}\right)^{2}} + \frac{x_{2}}{\left(1 - x_{2}\right)^{2}} + \frac{1}{1 - x_{2}}\right) \\\\
> > > \frac{x_{2}}{\left(1 - x_{2}\right)^{2}} + \frac{1}{1 - x_{2}} + \frac{x_{0}}{\left(1 - x_{0}\right)^{2}} + \frac{1}{1 - x_{0}} & 2 \left(\frac{1}{\left(1 - x_{1}\right)^{2}} + \frac{x_{2}}{\left(1 - x_{2}\right)^{2}} + \frac{1}{1 - x_{2}}\right) & \frac{2 \left(2x_{1} + x_{0}\right)}{\left(1 - x_{2}\right)^{3}}
> > > \end{bmatrix}$.
> > >
> > > Next, we will see that $f(\cdot)$ is $(\rho^c_k, \mu)$-RWC with $\rho_k^c$ defined as in Corollary 7.1, and $\mu(x) = \frac{1}{2}\Vert x \Vert^2$. It should be noted that the same proof below would have worked for the two other choices of $\mu(\cdot)$ suggested in Corollary 7.1 as well.
> > >
> > > Assume that $(x_1,x_2,x_3)^T \in \Delta_3^c$ for some choice of $c \in \left(0, \frac{1}{3}\right)$. It follows that the $\rho$ from Corollary 7.1 is:
> > >
> > > $\rho_k^c = c^{-2} n \left(k^2 + k - 1\right)  \cdot \max_i |\ell(C_i)| = c^{-2} \cdot 3 \cdot (4 + 2 - 1) \cdot 2 = 30 c^{-2}$.
> > >
> > > Please note that $\rho_k^c$ depends on $\max_i |\ell(C_i)|$. In terms of the theoretical convergence results, this is the main point in which the choice of $\ell(\cdot)$ comes into play.
> > >
> > > To prove that $f(\cdot)$ is $(\rho^c_k, \mu)$-RWC over $\Delta_3^c$, it is sufficient to show that:
> > >
> > > $\nabla^2 f(x) + \rho \nabla^2 \mu(x) \succeq 0$,
> > >
> > > which is equivalent, by our choice of $\mu$, to:
> > >
> > > $\nabla^2 f(x) + \rho I \succeq 0$.
> > >
> > > To prove that, it is sufficient to show that the resulting matrix on the left-hand side is diagonally dominant. In our case, since all elements of the Hessian are nonnegative for $x \in \Delta_3^c$, it is sufficient to show that:
> > >
> > > 1. $\rho^c_k \ge \frac{x_{2}}{\left(1 - x_{2}\right)^{2}} + \frac{1}{1 - x_{2}} + \frac{x_{0}}{\left(1 - x_{0}\right)^{2}} + \frac{1}{1 - x_{0}}$,
> > > 2. $\rho^c_k \ge 2 \left(\frac{1}{\left(1 - x_{1}\right)^{2}} + \frac{x_{2}}{\left(1 - x_{2}\right)^{2}} + \frac{1}{1 - x_{2}}\right)$,
> > > 3. $\rho^c_k \ge \frac{x_{0}}{\left(1 - x_{0}\right)^{2}} + \frac{1}{1 - x_{0}} + \frac{x_{2}}{\left(1 - x_{2}\right)^{2}} + \frac{1}{1 - x_{2}} + 2 \left(\frac{1}{\left(1 - x_{1}\right)^{2}} + \frac{x_{2}}{\left(1 - x_{2}\right)^{2}} + \frac{1}{1 - x_{2}}\right)$.
> > >
> > > In all these cases, it can be shown that the right-hand side of the inequalities is upper-bounded by $10c^{-2}$, which is smaller than $\rho_k^c$.
> > >
> > > Therefore, $f(\cdot)$, which was constructed from $\ell(\cdot)$, is $\rho_k^c$-RWC with respect to $\mu(\cdot)$. Corollary 7.1 shows that for every loss function $\ell(\cdot)$, one can construct an expectation loss function $f(\cdot)$ in this way, and that the resulting function $f(\cdot)$ is $(\rho,\mu)$-RWC with respect to the three options of $\mu(\cdot)$ outlined in the Corollary. It should be noted that the value of $\rho$ depends on $\ell(\cdot)$, as it is a function of $\max\limits_{C \in \mathcal{C}^k} \left|\ell(C)\right|$.

---

> > > > ### Comment · Reviewer_cTYy · 2024-11-27
> > > >
> > > > I see, thanks. I will retain my positive score

---

> > > > > ### Author Response · Authors · 2024-11-27
> > > > >
> > > > > Thank you.

---

### Author Response · Authors · 2024-11-20

Thank you for the positive and constructive feedback provided by all reviewers. We have worked to address your comments and suggestions through both this rebuttal and revisions to the manuscript. Below, we summarize the major changes and clarifications made in response to your feedback:

**Experimental Enhancements and Results**
We conducted additional synthetic experiments on more classical subset selection problems to better demonstrate the performance and robustness of our Subset Selection algorithm. These new experiments complement the Transfer Learning experiment, which remains in the main text, while the synthetic results have been added to the appendix.
The new synthetic experiments illustrate how favorable structure strongly influences the convergence rate of our algorithm. This behavior is tied to how informative the subset loss $\ell(C)$ is to the loss of other subsets that share elements with $C$.

**Theoretical Clarifications and Adjustments**
We revised the Mirror Descent step to use a more standard and equivalent notation, improving clarity.
The statements of Theorems 5.1 and 5.2 were updated to clarify that a decrease in $\tau$ leads to a decrease in $c$. A new remark was added to further elucidate this relationship.
We replaced mentions of the REINFORCE algorithm with alternative intuitions to better align with the context of our work.

**Subset Distribution and Sampling**
The distribution classes used in our method satisfy key criteria, including representation of optimal subsets, efficient gradient computation, and favorable optimization properties. We acknowledge that alternative distributions satisfying these criteria could achieve similar results and have flagged this as an area for future exploration.

**Other Clarifications**
We included a brief explanation of TL in Section 8 to better contextualize its relevance.


**Phrasing Modifications**
Some sentences were slightly modified to enhance conciseness in the purpose of freeing up space to accommodate for the revisions following the reviewers’ comments and suggestions. These modifications do not alter the content or intent of the paper.

---

### Meta-Review · Area_Chair_FtXA · 2024-12-09

**Metareview:**

This paper introduces a novel stochastic optimization framework for the subset selection problem The authors propose two innovative stochastic formulations within a black-box setting, where only subset metric values are accessible. By leveraging Stochastic Mirror Descent (SMD) to optimize the distribution parameters, the study provides significant theoretical contributions.

The paper is noted for its clear and well-organized presentation, strong theoretical foundations, and originality of approach. Methodological innovations and comprehensive theoretical analysis are identified as major strengths. Additionally, the empirical validation in transfer learning effectively demonstrates the practical relevance.

**Additional Comments On Reviewer Discussion:**

Reviewers provided constructive feedback highlighting areas for improvement, such as expanding empirical evaluations to include more standard subset selection tasks, comparing with existing deterministic and stochastic methods, justifying the choice of subset distributions, and discussing computational scalability. The authors addressed these comments by expanding their experiments, including comparisons with baseline methods, providing clearer explanations of their methodological choices, and discussing strategies for enhancing computational scalability.

---

### Decision · Program_Chairs · 2025-01-22

Accept (Poster)